

**Spatiotemporal variability and contribution of different aerosol**
**types to the Aerosol Optical Depth over the Eastern Mediterranean**
**Aristeidis K. Georgoulias[1,2,3,*], Georgia Alexandri[4,5], Konstantinos A. Kourtidis[5],**
**Jos Lelieveld[3,6], Prodromos Zanis[1], Ulrich Pöschl[2], Robert Levy[7], Vassilis**
**Amiridis[8], Eleni Marinou[4,8], Athanasios Tsikerdekis[1]**
[1] Department of Meteorology and Climatology, School of Geology, Aristotle University of
Thessaloniki, 54124, Thessaloniki, Greece
[2] Multiphase Chemistry Department, Max Planck Institute for Chemistry, D-55128, Mainz,
Germany
[3] Energy, Environment and Water Research Center, The Cyprus Institute, Nicosia, Cyprus
[4] Laboratory of Atmospheric Physics, Physics Department, Aristotle University of
Thessaloniki, 54124, Thessaloniki, Greece
[5] Laboratory of Atmospheric Pollution and Pollution Control Engineering of Atmospheric
Pollutants, Department of Environmental Engineering, Democritus University of Thrace,
67100, Xanthi, Greece
[6] Atmospheric Chemistry Department, Max Planck Institute for Chemistry, D-55128,
Mainz, Germany
[7] Earth Science Division, NASA Goddard Space Flight Center, MD 20771, Greenbelt, USA
[8] Institute for Astronomy, Astrophysics, Space Application and Remote Sensing, National
Observatory of Athens, 15236 Athens, Greece
*current address: Laboratory of Atmospheric Pollution and Pollution Control Engineering of
Atmospheric Pollutants, Department of Environmental Engineering, Democritus University of
Thrace, 67100, Xanthi, Greece
Correspondence to: A. K. Georgoulias (argeor@env.duth.gr)
**Abstract**
This study characterizes the spatiotemporal variability and relative contribution of different
types of aerosols to the Aerosol Optical Depth (AOD) over the Eastern Mediterranean as
derived from MODIS Terra (3/2000-12/2012) and Aqua (7/2002-12/2012) satellite
instruments. For this purpose, a 0.1° x 0.1° gridded MODIS dataset was compiled and





validated against AERONET sunphotometric observations. The high spatial resolution and
long temporal coverage of the dataset allows for the determination of local hot spots like
megacities, medium sized cities, industrial zones, and power plant complexes, seasonal
variabilities, and decadal averages. The average AOD at 550 nm ($AOD_{550}$) for the entire
region is ~ 0.22 ± 0.19 with maximum values in summer and seasonal variabilities that can be
attributed to precipitation, photochemical production of secondary organic aerosols, transport
of pollution and smoke from biomass burning in Central and Eastern Europe, and transport of
dust from the Sahara Desert and the Middle East. The MODIS data were analyzed together
with data from other satellite sensors, reanalysis projects and a chemistry-aerosol-transport
model using an optimized algorithm tailored for the region and capable of estimating the
contribution of different aerosol types to the total $AOD_{550}$. The spatial and temporal
variability of anthropogenic, dust and fine mode natural aerosols over land and anthropogenic,
dust and marine aerosols over the sea is examined. The relative contribution of the different
aerosol types to the total $AOD_{550}$ exhibits a low/high seasonal variability over land/oceanic
areas, respectively. Overall, anthropogenic aerosols, dust and fine mode natural aerosols
account for ~ 51 %, ~ 34 % and ~ 15 % of the total $AOD_{550}$ over land, while, anthropogenic
aerosols, dust and marine aerosols account ~ 40 %, ~ 34 % and ~ 26 % of the total $AOD_{550}$
over the sea, based on MODIS Terra and Aqua observations.

## 20 1 Introduction

For more than fifteen years, two MODIS (Moderate Resolution Imaging Spectroradiometer)
satellite sensors monitor tropospheric aerosols at a global scale on a daily basis. The retrieved
aerosol optical properties have been used in numerous air quality studies as well as studies
related to the effect of airborne particles on various climatic parameters (e.g. radiation, clouds,
precipitation, etc.). The $1^{o}$ x $1^{o}$ daily gridded level-3 dataset is primarily used for global as well
as regional studies while the single pixel level-2 data with a 10 km resolution (at nadir) are
mostly used for regional and local scale studies. Nevertheless, the use of the coarse resolution
MODIS data has predominated even in regional studies. The reasons for this could be the
smaller file size which makes their processing and storage easier or the fact that they are easily
accessible through user-friendly data bases which also allow for a very basic analysis like e.g.
NASA's GIOVANNI website (http://giovanni.gsfc.nasa.gov/giovanni/) (Acker and Leptoukh,

32 2007).





The same holds for studies focusing on the Mediterranean Basin, an area which is considered of
particular sensitivity as far as air pollution and climate change is concerned (Lelieveld et al.,
2002, Giorgi, 2006). The Mediterranean basin is one of the regions with highest aerosol optical
depths (AODs) in the world (Husar et al., 1997; Ichocku et al., 2005; Papadimas et al., 2008),
causing significant climate forcing especially in summer, which is characterized by low
cloudiness and high incoming solar radiation levels (Papadimas et al., 2012; Alexandri et al.,
2015). The Mediterranean is also recognized as a crossroads between three continents where
aerosols of various types accumulate. Marine aerosols from the Mediterranean Sea and even the
Atlantic Ocean combine with aerosols from continental Europe (urban and rural), dust particles
transported from the Sahara Desert and Middle East as well as biomass burning aerosols from
occasional wild fires and agricultural burning (Lelieveld et al., 2002). Specifically, as discussed
in Hatzianastassiou et al. (2009), the Eastern Mediterranean, the region under investigation here,
is located at a "key" point of this crossroads. There is a significant number of ground and
satellite-based studies on the abundance and optical properties of tropospheric aerosols in the
area; however, these studies are either focused on specific spots or applied a coarse spatial and
temporal resolution.
The ground-based instrumentation used in studies focusing on the aerosol load and optical
properties over the Eastern Mediterranean includes active and passive sensors such as Lidars
(e.g. Papayannis and Balis, 1998; Balis et al., 2004; Papayannis et al., 2005, 2009; Amiridis et
al., 2005, 2009; Mamouri et al., 2013; Kokkalis et al., 2013; Nisantzi et al., 2015), Cimel
sunphotometers (e.g. Israelevich et al., 2003; Kubilay et al., 2003; Derimian et al., 2006;
Kalivitis et al., 2007; Kelektsoglou and Rapsomanikis, 2011; Nikitidou and Kazantzidis, 2013),
Brewer spectrophotometers (e.g. Kazadzis et al., 2007; Koukouli et al., 2010), Multi-Filter
Radiometers (e.g. Gerasopoulos et al., 2009, 2011; Kazadzis et al., 2014), ceilometers (e.g.
Tsaknakis et al., 2011), Microtops sunphotometers (e.g. El-Metwally and Alfaro al., 2013), etc.
However, these and other studies not referenced here either refer to specific spots with the
majority of the ground stations being situated in large urban centers (e.g. Athens, Thessaloniki,
Cairo) or to specific events (e.g. Sahara dust intrusions, biomass burning events, etc.).
On the other hand, AOD and other aerosol optical properties have been studied over the greater
Eastern Mediterranean region based on data from Meteosat (Moulin et al., 1998), SeaWIFS
(Koren et al., 2003; Antoine and Nobileau, 2006; Mélin et al., 2007; Nabat et al., 2013), TOMS
(Alpert and Ganor, 2001; Israelevich et al., 2002; Koukouli et al. 2006; Hatzianastassiou et al.,
2009; Koukouli et al., 2010, Israelevich et al., 2012, Kaskaoutis et al., 2012a; Nabat et al., 2013;





Gkikas et al., 2013; 2014; Varga et al., 2014), MODIS Terra and Aqua (Barnaba and Gobbi,
2004; Papayannis et al., 2005; Kaskaoutis et al., 2007; 2008; 2010; 2011; 2012a,b,c,d;
Kosmopoulos et al., 2008; Papadimas et al., 2008, 2009; Rudich et al., 2008; Carmona and
Alpert, 2009; Karnieli et al., 2009; Gkikas et al., 2009; 2013; Hatzianastassiou et al., 2009; El-
Metwally et al., 2010; Koukouli et al., 2010; Kanakidou et al., 2011; Gerasopoulos et al., 2011;
de Meij and Lelieveld, 2011; Marey et al., 2011; de Meij et al., 2012; Nabat et al., 2012, 2013;
Nikitidou and Kazantzidis, 2013; Athanasiou et al., 2013; Benas et al., 2011; 2013; Sorek-
Hamer et al., 2013; Kabatas et al., 2014; Kourtidis et al., 2014; Mishra et al., 2014; Flaounas et
al., 2015; Kloog et al., 2015), OMI/AURA (Kaskaoutis et al., 2010; El-Metwally et al., 2010;
Marey et al., 2011; Kaskaoutis et al., 2012b,c, Gkikas et al., 2013; 2014; Sorek-Hamer et al.,
2013; Varga et al., 2014; Flaounas et al., 2015), CALIOP/CALIPSO (Amiridis et al., 2009,
2013, Mamouri et al., 2009; Marey et al., 2011; Kaskaoutis et al., 2012c; de Meij et al., 2012;
Nabat et al., 2012, 2013; Mamouri and Ansmann, 2015), MISR/Terra (Kanakidou et al., 2011;
Marey et al., 2011; de Meij and Lelieveld, 2011; de Meij et al., 2012; Nabat et al., 2013;
Kabatas et al., 2014; Abdelkader et al., 2015) as well as NOAA/AVHRR, MERIS/ENVISAT,
AATSR/ENVISAT, PARASOL/POLDER, MSG/SEVIRI, and Landsat satellite data (see
Retalis and Sifakis, 2010; Nabat et al., 2013; Benas et al., 2013; Sifakis et al., 2014). To our
knowledge, these studies comprise the majority of work focusing on tropospheric aerosols over
the Eastern Mediterranean by means of satellite remote sensing, published in peer reviewed
journals the last ~ 15 years. As shown in Fig. 4 of this work, the publication rate of satellite-
based studies focusing on Eastern Mediterranean aerosols nearly doubled every three years
during the period 1997-2014 which is indicative of the increasing scientific interest in the area.
In a very large fraction of the satellite-based studies referenced above, the used data are either of
coarse mode (usually $1^{o}$ which is ~ 100 km for the mid-latitudes) or focus on specific spots for
validation purposes. In a few cases, high resolution data were used in spatiotemporal studies;
however, either these studies are restricted over surfaces covered by water or examine a short
period only. For example, Moulin et al. (1998) investigated the dust AOD patterns over the
oceanic areas of the Mediterranean Basin at a resolution of 35 x 35 $km^2$ for a period of 11 years
(1984-1994) using Meteosat observations. A 7-year climatology (1998-2004) of total and dust
AOD for the same regions at a resolution of $0.16^{o}$ x $0.16^{o}$ was compiled by Antoine and
Nobileau (2006) using observations from SeaWIFS. Mélin et al. (2007) merged SeaWIFS and
MODIS data and presented high resolution AOD patterns (2 x 2 $km^2$) for May 2003. As far as
MODIS is concerned, only Barnaba and Gobbi (2004) presented a high resolution ($0.1^{o}$ x $0.1^{o}$)





spatiotemporal analysis for a period of 1 year (2001) over ocean only. In a recent paper,
Athanasiou et al. (2013) presented in detail a method for compiling a 0.5-degree resolution
AOD gridded dataset using level-2 MODIS Terra data for the greater region of Greece (2000-
2008). However, the spatial resolution they used (~ 50 km) is not high enough to reveal local
sources (e.g. cities, islands, river banks, etc.). Overall, there has not been so far any detailed
high resolution spatiotemporal study of the AOD over Eastern Mediterranean.
In this paper, the $AOD_{550}$ spatiotemporal variability over the Eastern Mediterranean ($30^{o}$N-
$45^{o}$N, $17.5^{o}$E-$37.5^{o}$E) is presented at a high spatial resolution ($0.1^{o}$ x $0.1^{o}$) based on MODIS
Terra and Aqua observations. Level-2 MODIS data are used for the compilation of a 0.1-degree
gridded dataset which is validated against ground-based observations. In order to calculate the
contribution of different aerosol types to the total AOD, the MODIS data were analyzed
together with other satellite data, ERA-Interim reanalysis data and the Goddard Chemistry
Aerosol Radiation and Transport (GOCART) model using an algorithm optimized for the
surface properties of the Eastern Mediterranean region. The different datasets used in this
research are presented in detail in Sect. 2 while a detailed description of the method is given in
Sect. 3. Sect. 4 includes the results from the MODIS validation procedure, the annual and
seasonal variability of $AOD_{550}$ over the region with a discussion on the local aerosol sources
and the differences between Terra and Aqua, and the annual and seasonal contribution of
different aerosol types to the total $AOD_{550}$. Finally, in Sect. 5, the main conclusions of the paper
are presented along with a short discussion on how these results could contribute to future
studies in the area.

## 2    Observations, reanalysis data and model simulations

### 2.1    MODIS Terra and Aqua satellite observations

The main data used in this work come from the level-2 MODIS Terra (MOD04_L2) and
MODIS Aqua (MYD04_L2) Collection 051 dataset and have been acquired through NASA's
Level 1 and Atmosphere Archive and Distribution System (LAADS)
(http://ladsweb.nascom.nasa.gov). The fact that MODIS Terra and Aqua have a daytime
equator crossing time at 10:30 LT (morning) and 13:30 LT (noon), respectively. MODIS
instruments with a viewing swath of 2330 km measure backscattered radiation at 36 spectral
bands between 0.415 and 14.235 μm with a spatial resolution of 250, 500 and 1000 m,
providing a nearly global coverage on a daily basis. Aerosol optical properties for the
standard MODIS aerosol product are retrieved using two different "Dark Target" (DT)





algorithms. The one is used over land surfaces (Kaufman et al., 1997; Levy et al., 2007a, b;
Remer et al., 2005; Levy et al., 2010) and the other over oceanic regions (Tanré et al., 1997;
Levy et al., 2003; Remer et al., 2005). The "Deep Blue" algorithm (DB) (Hsu et al., 2004;
Hsu et al., 2006) has been used for retrievals over bright land surfaces (e.g. deserts) where the
DT algorithm fails. Only recently, updates to the algorithm allowed for extending the spatial
coverage of the DB aerosol product over all land areas (Hsu et al., 2013; Sayer et al., 2013;
2014). AERONET Cimel sunphotometric measurements have been extensively used for the
validation of the MODIS over-land and over-ocean products (e.g. Chu et al., 2002; Remer et
al., 2002; Remer et al., 2005; Levy et al., 2010; Shi et al., 2013).
In this work, $AOD_{550}$ over both land and ocean and the Fine Mode Ratio ($FMR_{550}$) over ocean
from Collection 051 were used at a spatial resolution of 10 km (at nadir). The uncertainty of
the MODIS aerosol optical depth has been estimated at ±(0.05+0.15AOD) over land (Chu et
al., 2002; Levy et al., 2010) and ±(0.03+0.05AOD) over ocean (Remer et al., 2002) relative to
the AERONET AOD. Specifically, for the DT data used in this work only high quality
retrievals are used over land. This means that the data have a Quality Assurance Confidence
(QAC) flag equal to 3 (high confidence). For retrievals over ocean we use data with a QAC
flag of 1 (marginally good), 2 (good) and 3 (see Levy et al., 2009 for details). The pre-launch
uncertainty of $FMR_{550}$ is ±30  % over ocean (Remer et al., 2005) while over land this
parameter is by no means trustworthy and should only be used in qualitative studies (e.g. see
Georgoulias and Kourtidis, 2011). In cases where DT algorithm does not provide products
over land, especially over bright arid and semi-arid regions of North Africa, $AOD_{550}$ values
from the DB algorithm are used in our work. The expected uncertainty of the DB product
used here is ±(0.05+0.2AOD) relative to the AERONET AOD (Hsu et al., 2006). The
analyzed datasets cover the period from 3/2000 to 12/2012 for Terra and from 7/2002 to
12/2012 for Aqua MODIS covering the region of Eastern Mediterranean. The Collection 051
DB data for Terra are available only until 12/2007 due to calibration issues; nevertheless,
these data are carefully used within our analysis to get a complete image of the aerosol load
over the region.
**2.2   AERONET ground-based observations**
For the evaluation of the MODIS $AOD_{550}$, data from 13 AERONET Cimel network ground
stations   in   the   region   of   Eastern   Mediterranean   have   been   acquired
(http://aeronet.gsfc.nasa.gov). The stations were selected such as their operation period covers





at least 2 years and there are at least 100 common days of co-localized AERONET and
MODIS observations. AERONET Cimel sunphotometers measure solar radiation every 15
minutes within the spectral range from 340 to 1020 nm (Holben et al., 2001). The spectral
measurements allow for the retrieval of columnar aerosol properties (see Holben et al., 1998;
Dubovik and King, 2000; Dubovik et al., 2000, 2002). The AERONET AOD uncertainty is in
the order of 0.01-0.02 (Eck et al., 1999), being larger at shorter wavelengths. Here, we use
quadratic fits on a log-log scale to interpolate the AERONET data (AODs at 440, 500, 675
and 870 nm) to the MODIS band-effective wavelength of 550 nm (Eck et al., 1999; Levy et
al., 2010). Simultaneous measurements of the Ångström Exponent (AE) for the spectral range
440-870 nm (AE440-870) from the 13 AERONET stations mentioned above were also
utilized in this work in order to account for days with dust dominance. The uncertainty of the
AE is significantly higher than the AOD uncertainty, especially under low-AOD conditions.
Li et al. (2014) found that the uncertainty for a typical Northern Hemispheric AERONET
station (GSFC) is ~ 0.6 during winter when AODs are significantly lower compared to
summer (~ 0.15).

## 17    2.3   LIVAS CALIOP/CALIPSO dust climatology

Dust aerosol optical depths at 532 nm ($AOD_{532}$) from CALIOP/CALIPSO (Cloud-Aerosol
Lidar with Orthogonal Polarization instrument aboard Cloud-Aerosol Lidar and Infrared
Pathfinder Satellite Observations satellite) at a resolution of $1^o$ x $1^o$ are also used here for the
period 2007-2012. CALIPSO measures cloud and aerosol properties flying at a 705 km sun
synchronous polar orbit with a 16 day repeat cycle and an equator-crossing time close to that
of the Aqua satellite (13:30 LT). The dust product used here comes from a Saharan-dust-
optimized retrieval scheme that was developed within the framework of the LIVAS (Lidar
Climatology of Vertical Aerosol Structure for Space-Based LIDAR Simulation Studies)
project (Amiridis et al. 2015) and has been presented in detail in Amiridis et al. (2013). In
brief, the LIVAS dust product is optimized for Europe by using a lidar ratio of 58 sr instead of
40 sr is applied to Level 2 dust related backscatter products. This correction results to an
improved $AOD_{532}$ absolute bias of ~ -0.03 compared to spatially and temporally co-located
AERONET observations (Amiridis et al., 2013). The corresponding reported biases for the
original CALIPSO data are significantly higher (~ -0.10). The bias is even lower when
compared against MODIS satellite-based observations. Other improvements of this product
are related to the use of a new methodology for the calculation of pure dust extinction from





dust mixtures and the application of an averaging scheme that includes zero extinction values
for the non-dust aerosol types detected. Overall, this product (hereafter denoted as LIVAS
dust product) exhibits better agreement with observations from MODIS and AERONET and
simulations from the BSC-DREAM8b dust model over North Africa and Europe than the
standard CALIPSO data hence being an ideal tool for the evaluation of other satellite-based
products.
**2.4   Earth Probe TOMS and OMI satellite observations**
For this work, Aerosol Index (AI) data (Herman et al., 1997) from the Earth Probe TOMS
(Total Ozone Mapping) spectrometer aboard Earth Probe for the period 1/2000-9/2004 at a
resolution of $1^{o}$ x $1.25^{o}$ and the OMI (Ozone Monitoring Instrument) sensor aboard EOS
AURA for the period 10/2004-12/2012 at a resolution of $1^{o}$ x $1^{o}$ were acquired through the
GIOVANNI web database (http://giovanni.gsfc.nasa.gov/giovanni/). Earth Probe TOMS
continued the record of the first three TOMS instruments aboard Nimbus-7, Meteor-3 and
ADEOS flying in a sun synchronous orbit at an altitude of 740 km with an instantaneous field
of view size of 39 x 39 $km^2$ at nadir. The instrument had an ascending node equator crossing
time at 12:00 LT covering 85 % of the globe on a daily basis from 7/1996 until 12/2005. The
satellite was originally set to a 500 km sun synchronous orbit but was set to its final orbit after
the failure of ADEOS satellite in 6/1997. OMI is a UV/VIS nadir solar backscatter
spectrometer (Levelt et al., 2006) that continues the long TOMS record. OMI flies in a sun
synchronous polar orbit at an altitude of 705 km with an ascending node equator crossing
time at 13:45 LT. Its 2600 km viewing swath allows for almost daily global coverage while
the spatial resolution of the instrument is 13 x 24 $km^2$ at nadir. The AI which is calculated by
the two instruments constitutes a qualitative indicator of the presence of UV absorbing
aerosols in the atmosphere (e.g. biomass burning, dust). The Version 8 algorithm is applied on
spectral measurements from both TOMS and OMI sensor to produce a consistent long-term
AI timeseries (Li et al., 2009). AI is calculated from the difference in surface reflectivity
derived from the 331.2 and 360 nm measurements exhibiting an uncertainty of ±30 % (Torres
et al., 2007).
**2.5   ERA-Interim reanalysis data**
Wind speed (ws) data at 10 m above surface from the ERA-Interim reanalysis (Dee et al.,
2011) are used for 9:00 and 12:00 UTC on a daily basis for the period 2000-2012. We use





9:00 and 12:00 UTC data in order to be closer to the Terra and Aqua overpass time in the
area, respectively. The various ERA-Interim reanalysis fields are produced by ECMWF's
Integrated Forecast System (IFS) assimilating satellite and ground-based observations. The
system includes a 4-D variational analysis with a 12-hour analysis window. The spatial
resolution of the ERA-Interim data is ~ 79 km with 60 vertical levels from the surface up to
0.1 hPa while the data can be acquired at various resolutions through ECMWF's website
(http://apps.ecmwf.int/datasets/data/interim-full-daily/). Over ocean, the 10 m ERA-interim
wind speed exhibits a bias of less than -0.5 m/s compared to quality-controlled in situ
observations on a global scale (Dee et al., 2011). Specifically, for the region of Eastern
Mediterranean examined here, the 10 m ERA-interim wind speed exhibits a bias of -0.96 m/s
(-16 %) compared to satellite-based observations from QuikSCAT (Hermann et al., 2011).
**2.6   MACC reanalysis data**
The daily MACC total and dust $AOD_{550}$ data for the period 2003-2012 come from the aerosol
analysis and forecast system of ECMWF which consists of a forward model (Morcrette et al.,
2009) and a data-assimilation module (Benedetti et al., 2009). $AOD_{550}$ measurements from
the two MODIS instruments aboard Terra and Aqua are assimilated by the MACC forecasting
system through a 4D-Var assimilation algorithm to produce the aerosol analysis, leading to an
improved AOD representation compared to observations (see Benedetti et al., 2009; Mangold
et al., 2011). Five aerosol species are included within MACC, namely, mineral dust, sea salt,
sulfates, black carbon and organic matter. Three different size bins are used for mineral dust
and sea salt particles while the black carbon and organic material are distributed to a
hydrophilic and a hydrophobic mode. Dust and sea salt emissions are given as a function of
surface wind speed, while the emissions of the other species are taken from inventories. The
spatial resolution of the MACC reanalysis data is ~ 79 km with 60 vertical levels from the
surface up to 0.1 hPa and can be acquired through: http://apps.ecmwf.int/datasets/data/macc-
reanalysis/. The MACC total and dust AODs have been evaluated against ground and
satellite-based observations (see Elguindi et al., 2010; Bellouin et al., 2013; Inness et al.,
2013; Cesnulyte et al., 2014; Cuevas et al., 2015) showing that the MACC aerosol products
generally capture well the daily, seasonal and interannual variability of aerosols. As discussed
in Bellouin et al. (2013) the uncertainties of MACC total $AOD_{550}$ (~ 0.03) and dust $AOD_{550}$ (~
0.014) arise from uncertainties in the MODIS retrievals which are assimilated into the model
and errors in the forward modeling of total and component AODs.



## 2.7  GOCART data

Daily total and dust $AOD_{550}$ data from the GOCART chemistry-aerosol-transport model
simulations (version 006) are used in this study for the period 2000-2007. The GOCART
model (see Chin et al., 2000, 2002, 2004, 2007; Ginoux et al., 2001, 2004) uses the
assimilated meteorological fields of the Goddard Earth Observing System Data Assimilation
System (GEOS DAS) which are generated by the Goddard Global Modeling and Assimilation
Office (GMAO). The data which are used were acquired from an older version of NASA's
GIOVANNI web database (http://disc.sci.gsfc.nasa.gov/giovanni/) and come from a
simulation implemented at a spatial resolution of 2° (latitude) x 2.5° (longitude) with 30
vertical sigma layers (Chin et al., 2009). The model includes physicochemical processes of
major tropospheric aerosol components (sulfates, dust, black carbon, organic carbon, sea salt)
and precursor gases ($SO_2$ and dimethylsulfide) incorporating various atmospheric processes.
The total $AOD_{550}$ from GOCART compared to ground-based observations from the
AERONET exhibits a relative mean bias [mean(GOCART)/mean(AERONET)] of 1.120,
1.135 and 0.959 over Europe, North Africa and for the whole globe, respectively.

## 2.8  Ancillary data

Apart from the primary datasets presented above, three additional datasets were used in order
to support our findings. OMI/AURA daily gridded (Bucsela et al., 2013) tropospheric $NO_2$
columnar data (OMNO2d version 2.1) at a spatial resolution of $0.25^{o}$ x $0.25^{o}$ were acquired
from NASA's GIOVANNI web database (http://giovanni.gsfc.nasa.gov/giovanni/) for the
period 2005-2012. The quality checked data used in this work correspond to sky conditions
where cloud fraction is less than 30 %. Planetary boundary layer (PBL) $SO_2$ daily gridded
columnar data (OMSO2e version 1.1.7) were also acquired from GIOVANNI for the same
period. The OMSO2e gridded data ($0.25^{o}$ x $0.25^{o}$) used in this work are produced from best
level-2 pixel data, screened for OMI row anomaly and other data quality flags. The PBL $SO_2$
column retrievals are produced with an algorithm based on principal component analysis
(PCA) of the OMI radiance data (Li et al., 2013). Finally, monthly precipitation data from the
3B43 TRMM and Other Sources Monthly Rainfall Product (version 7) at a spatial resolution
of $0.25^{o}$ x $0.25^{o}$ for the period 2000-2012 were obtained from GIOVANNI. This dataset is
derived from 3-hourly precipitation retrievals from the Precipitation Radar (PR), the TRMM
Microwave Imager (TMI) and the Visible and Infrared Scanner (VIRS) aboard the TRMM





(Tropical Rainfall Monitoring Mission) satellite merged with other satellite-based
precipitation data and the Global Precipitation Climatology Centre (GPCC) rain gauge
analysis (Huffman et al., 2007).
**3    Methodology**
**3.1    Compiling a MODIS 0.1$^{o}$ x 0.1$^{o}$ gridded dataset**
To investigate the spatial and temporal variability of aerosols over the Eastern Mediterranean
we first created a 0.1$^{o}$ x 0.1$^{o}$ daily gridded aerosol dataset using single pixel level-2 $AOD_{550}$ and
$FMR_{550}$ data from MODIS Collection 051. The same resolution has been utilized in previous
studies (e.g. Barnaba and Gobbi, 2004) in the region; however, without reporting on the
gridding methodology followed. In this work we present a gridding methodology that could be
used as a reference for future regional studies. The methodology has been successfully applied
in the past on level-2 MODIS Terra data in different cases studies, e.g. in order to examine the
weekly cycle patterns of $AOD_{550}$ over the region of Central Europe and the aerosol load
changes observed over a cement plant in Greece due to changes in the deposition practices of
the primary materials (see Georgoulias and Kourtidis, 2012; Georgoulias et al., 2012; Kourtidis
et al., 2014). In the following lines we proceed to a detailed description of the method
underlining the potential of being used in detailed quantitative studies like this one.
First, a 0.1$^{o}$ x 0.1$^{o}$ resolution grid covering the Eastern Mediterranean (30$^{o}$N-45$^{o}$N, 17.5$^{o}$E-
37.5$^{o}$E) is defined which corresponds to 30000 grid cells. As already mentioned in Sect. 2.1,
only level-2 single pixel $AOD_{550}$ measurements with a QAC flag of 3 and a QAC flag greater
than 0 were used over land and over ocean, respectively, to ensure the high quality of the data.
Pixels are attributed at a specific grid cell if their center falls within a 25 x 25 km$^{2}$ square
window around the grid cell (see Fig. S1 in the Supplement). These pixels are then used for the
calculation of daily averages. As shown in Figure S1, a grid cell of 0.1$^{o}$ (~ 10 km) is as big as
the centre of a large Mediterranean city like Thessaloniki, Northern Greece (~ 1 million
inhabitants). The procedure was followed separately for MODIS Terra and Aqua data. In cases
of grid cells with no DT MODIS observations, data from the DB algorithm were used (over
bright arid and semi-arid regions of North Africa) constituting only a small part of the gridded
dataset.
The size of the gridding window was selected following Koukouli et al. (2007). They used both
10 and 25-km windows showing that the latter allows for the inclusion of more data points
without undermining the ability of monitoring accurately the aerosol load over a specific spot.



In addition, in cases of urban sites, a window of 25 km allows for the inclusion of pixels from
the surrounding non-urban surfaces where the MODIS surface reflectance parameterization is
better (Levy et al., 2010). The size of each MODIS pixel is 10 km at nadir, but at the swath
edges, it may become 2-3 times larger. Hence, ideally the maximum number of pixels that could
be used in the daily averaging is nine. The overlap between the windows of neighbouring grid
cells does not affect the representativeness of the dataset over each grid cell. Aerosols are
transported by air masses throughout the day and thus the aerosol load in neighbouring grid
cells is not expected to be completely independent.
In order to make sure that the use of a 25-km gridding window is optimal for capturing local
pollution sources we repeated the same procedure for bigger gridding windows (50-km, 75km
and 100-km) using MODIS Terra $AOD_{550}$ data for the year 2004. Numerous aerosol hot spots
cannot be seen as the gridding window becomes bigger and there is a significant smoothing of
the aerosol patterns mainly over land (Fig. S2). The use of the MODIS gridded dataset in the
detection of local aerosol hot spots is discussed in more detail in Sect. 4. In addition, we
conducted a detailed validation of the MODIS data against sunphotometric data from a total of
13 AERONET stations in the region (see Fig. 1). The validation procedure was repeated several
times for different spatial collocation windows which were equal to the windows used for the
gridding procedure (i.e. 25, 50, 75 and 100-km) and for different data quality criteria. The
results of the validation procedure are presented in Sect. 4.1 while part of them is given in the
Supplement of this manuscript (see Table S2). Overall, it is shown that the gridding
methodology followed here offers the best compromise for studying the spatial variability of
aerosols on a regional or local scale, preserving at the same time the representativeness of the
real aerosol load over each specific spot.
In order to generalize our results, nine different sub-regions (Fig. 1) were selected apart from
the three basic regions of interest, namely, the whole Eastern Mediterranean (EMT) and the land
(EML) and oceanic (EMO) areas of the region. The selection was done mainly taking into
account geographical but also land type and land use criteria. The four sub-regions that
correspond to the land regions of Eastern Mediterranean are the Northern Balkans Land (NBL),
the Southern Balkans Land (SBL), the Anatolia Land (ANL) and the Northern Africa Land
(NAL) region while the five sub-regions that correspond to the oceanic regions are the Black
Sea Oceanic (BSO), the North-Western Oceanic (NWO), the South-Western Oceanic (SWO),
the North-Eastern Oceanic (NEO) and the South-Eastern Oceanic (SEO) region. Mean values of





the total $AOD_{550}$ from the Terra and Aqua MODIS are reported for each one of the three basic
regions of interest and their nine sub-regions in Sect. 4.
### 3.2   Contribution of different aerosol types to $AOD_{550}$
### 3.2.1   Ocean
In order to quantify the contribution of different types of aerosols to the total $AOD_{550}$ we
followed a different approach for ocean and land regions. This is due to the lack of reliable
$FMR_{550}$ retrievals over land (e.g. see Levy et al., 2010; Georgoulias and Kourtidis, 2011) which
are crucial for the algorithms used in this work. Over ocean we utilize wind speed data at 10 m
above surface from the ERA-Interim reanalysis, AI data from TOMS and OMI along with
$AOD_{550}$ and $FMR_{550}$ from the MODIS Terra and Aqua gridded datasets presented above. All the
datasets were brought to the same 0.1 degree spatial resolution as MODIS by using bilinear
interpolation. In the case of TOMS and OMI we used monthly mean AI data following Bellouin
et al. (2008) in order to avoid gaps especially during the TOMS period.
In general, the algorithm used over oceanic regions (see Fig. 2) is similar with the one presented
in Bellouin et al. (2008). First, the marine $AOD_{550}$ ($\tau_m$) is calculated from near surface wind
speed using a linear relation which has been obtained from ground-based studies over pollution
free oceanic regions. Bellouin et al. (2008) use the linear relation of Smirnov (2003). Then, if $\tau_m$
is greater or equal than $AOD_{550}$ it is assumed that there are marine particles only over this
region. If $\tau_m$ is smaller than $AOD_{550}$ a decision tree is followed which is first based on $FMR_{550}$
and then on AI in order to reach a conclusion about the type of aerosols that account for
$AOD_{550}$. If $FMR_{550}$ is smaller than the critical value of 0.35 and AI is greater than or equal to a
critical value it is assumed that there are both marine aerosols ($\tau_m$) and dust ($\tau_d=AOD_{550}-\tau_m$)
while if AI is smaller than this critical value it is assumed that there are marine aerosols only.
The AI critical value is equal to 1 in Bellouin et al. (2008). If $FMR_{550}$ is greater than or equal to
0.83 it is assumed that there are both anthropogenic ($\tau_a=AOD_{550}-\tau_m$) and marine aerosols ($\tau_m$).
In the occasion of having a $FMR_{550}$ equal to 0.35 or greater than 0.35 but smaller than 0.83 one
has to take again AI into consideration. If AI is less than the critical value it is assumed that
there are marine aerosols ($\tau_m$) only while in the opposite occasion it is assumed that all the three
types of aerosols that can be defined over oceanic regions by this algorithm, namely, dust
$[\tau_d=(1-FMR_{550})(AOD_{550}-\tau_m)]$, anthropogenic $[\tau_a=FMR_{550}(AOD_{550}-\tau_m)]$ and marine aerosols ($\tau_m$)
are present. One should keep in mind that all the biomass burning aerosols are classified as
anthropogenic by this method.



In this work, we proceeded to a "fine-tuning" of the algorithm for the region of Eastern
Mediterranean. First, we applied the algorithm on MODIS Terra data using the same critical
values as in Bellouin et al. (2008). The results showed that the original Bellouin et al. (2008)
method might be valid for global studies but for a "closed" sea like the Mediterranean the use of
the original critical values leads to a large overestimation of sea salt AODs and therefore
underestimation of dust and anthropogenic aerosol AODs. Indicative of this situation is Fig. S3
in the Supplement where we present the relative contribution of dust, marine and anthropogenic
aerosols per month over the oceanic regions of Eastern Mediterranean as calculated using the
original Bellouin et al. (2008) method. It is shown that the marine contribution is several times
higher than the values reported for the Mediterranean Basin in previous studies (e.g. see Nabat
et al., 2012). Evaluation of the algorithm was done using dust $AOD_{532}$ data from the LIVAS
CALIOP/CALIPSO product. From LIVAS we only use the high quality Sahara dust product as
a reference and not other aerosol type retrievals (e.g. marine aerosols) since the dust retrievals
from CALIOP/CALIPSO are by far the most reliable (e.g. Burton et al., 2013). We performed
several tests by changing the linear relation that connects $\tau_m$ with near surface wind speed and
the AI critical values and compared each time the dust $AOD_{550}$ seasonal variability with the
LIVAS $AOD_{532}$ seasonal variability for the ocean covered sub-regions of Eastern
Mediterranean. Results from this algorithm-tuning procedure can be found in Figs. S4e-i of the
Supplement where one can also see the underestimation of dust $AOD_{550}$ from the original
Bellouin et al. (2008) algorithm.
The linear relation given in Kaufman et al. (2005) was finally selected ($\tau_m$=0.007ws+0.02). The
2000-2012 average wind speed over the ocean for the region of Eastern Mediterranean is ~ 5.3
m/s. Kaufman et al. (2005) reduced the offset in the linear relation of Smirnov (2003) from 0.06
to 0.02 to fit the average baseline AOD of 0.06 for the typical wind speed of 6 m/s. In addition,
our tests showed that a AI critical value of 0.5 performs better over the region of Eastern
Mediterranean. As discussed in Jones and Christopher (2011), the AI threshold of 0.5 is capable
of separating sea salt from UV absorbing aerosols (dust and biomass burning) efficiently.
Another test, following the example of other studies (see Lehahn et al., 2010), was to assume
that for wind speed less than 5 m/s there is very little or no sea-spray particle production
(limited bursting of entrained air bubbles associated with whitecap formation). In this case, $\tau_m$ is
stable, equal to the offset of the linear relation between $\tau_m$ and wind speed which is indicative of
the background sea salt $AOD_{550}$. However, this test reveals that the effect of assuming stable $\tau_m$
for wind speed less than 5 m/s is insignificant and therefore we selected to follow the Kaufman



et al. (2005) linear relationship for the whole wind speed range. As shown in Figs. S4e-i, the
seasonal variability when applying our modified algorithm over oceanic regions is very close to
the LIVAS dust $AOD_{532}$ especially for the months with lower dust load (June-January). It is also
shown that dust AODs from this algorithm are closer to the LIVAS dust product than dust
AODs from MACC reanalysis do. The slight overestimation of dust AOD or the shift of the
maximum dust load we observe for the period of high dust loads in the region (February-May)
is probably connected to the narrow swath and the 16-day time of CALIPSO which means that
several dust events might be not observed by the CALIOP instrument contrary to MODIS which
has a daily coverage.

### 11  3.2.2  Land

As already mentioned in the previous paragraph a different approach is followed over the land
regions of Eastern Mediterranean due the low confidence on the MODIS $FMR_{550}$ and Ångström
exponent retrievals over land compared to that over ocean (e.g. see Levy et al., 2010;
Georgoulias and Kourtidis, 2011). This limitation does not allow us to distinguish the
contribution of fine and coarse mode aerosols in terms of $AOD_{550}$. In this case, we choose to use
daily model fields of the dust contribution to the total AOD (here MACC reanalysis and
GOCART). We follow a method similar with the one presented in Bellouin et al. (2013).
Specifically, we calculate the dust $AOD_{550}$ by scaling the MODIS $AOD_{550}$ data with the MACC
or GOCART dust/total $AOD_{550}$ ratios [$f_d = \tau_{d(model)}/\tau_{(model)}$] on a daily basis.
Since the MACC data are available only from 2003 to 2012, in order to take advantage of the
full MODIS dataset (3/2000-12/2012), data from the GOCART model were used for the period
2000-2002. The GOCART data were normalized in order to be consistent with the MACC data.
Daily dust/total $AOD_{550}$ ratios ($f_d$) from the common GOCART-MACC period 2003-2007 were
first brought to a common $1^o$ x $1^o$ spatial resolution using bilinear interpolation and then we
calculated the regression line for each grid cell on a seasonal basis. The linear relations were
afterwards used in order to normalize the 2000-2002 GOCART ratios to have a homogeneous
dataset. The slopes and offsets of these regression lines and the corresponding correlation
coefficients (R) can be seen in Figs. S5, S6 and S7 of the Supplement, respectively. Overall, for
the whole time period, the MACC reanalysis $f_d$ ratios are lower by ~ 26 % from the GOCART $f_d$
ratios and the linear relation connecting the two products is $f_{dMACC} = 0.4964 f_{dGOCART} + 0.0952$
with a correlation coefficient R of 0.74. The $f_d$ values of the merged GOCART-MACC (2000-
2012) timeseries were checked using the Standard Normalized Homogeneity Test (SNHT) as



described in Alexandersson (1986). The statistical significance was checked following Khaliq
and Ouarda (2007) and the $f_d$ timeseries were found to be homogeneous (see Fig. S8 of the
Supplement). Hence, this test verifies that the use of the merged GOCART-MACC $f_d$ dataset
will not insert any artifacts (e.g. trends or breaks) in the algorithm. Finally, the $f_d$ data were
brought to the same spatial resolution with MODIS data ($0.1^o$ x$0.1^o$) using bilinear interpolation
again.
After the calculation of $\tau_d$ with the use of $f_d$ values ($\tau_d=f_d AOD_{550}$), we proceed to the calculation
of the anthropogenic contribution to the total $AOD_{550}$ ($\tau_a$) by multiplying the non-dust part of
$AOD_{550}$ with the anthropogenic fraction $f_a$ for the region of Eurasia (0.77±0.20) given in
Bellouin et al. (2013) [$\tau_a=f_a(1-f_d)AOD_{550}$]. The rest of the total $AOD_{550}$ is attributed to the fine
mode natural aerosols [$\tau_n=(1-f_a)(1-f_d)AOD_{550}$] (see Fig. 2). As discussed in Bellouin et al.
(2013), the fine mode natural aerosols consist of sea salt, dimethyl sulfide from land and
oceanic sources, $SO_2$ from degassing volcanoes and secondary organic aerosols from biogenic
emissions. It has to be highlighted that like in the case of oceanic regions the biomass burning
aerosols are classified as anthropogenic by this algorithm. As shown in Figs. S4a-d, the seasonal
variability of $\tau_d$ over land covered regions is very close to the LIVAS dust $AOD_{532}$ which is
used as a reference.
Overall, the algorithm described above performs well as far as dust is concerned. This is further
shown when comparing MODIS Terra and Aqua $\tau_d$ values with collocated AERONET
observations for dust dominated days (see Fig. S9 in the Supplement). The method followed for
the collocation of the data is similar to the one presented in Sect. 4.1 while dust dominated days
were days with an AERONET AE smaller than 1 (see Mateos et al., 2014) and a MODIS based
$\tau_d$ greater than $\tau_a$ and $\tau_n$ or $\tau_m$. The uncertainties of the calculated $\tau_a$, $\tau_d$, $\tau_n$ and $\tau_m$ values which
are inserted by the input data and the assumptions of the algorithm are expected to be similar
with the ones presented in Bellouin et al. (2013). Bellouin et al. (2013) using a Monte-Carlo
analysis indicated that $\tau_a$ can be specified with an uncertainty of ~ 23 % over land and ~ 16 %
over the ocean, $\tau_d$ can be specified with an uncertainty of ~ 19 % over land and ~ 33 % over the
ocean, $\tau_n$ can be specified with an uncertainty of ~ 41 % and $\tau_m$ with an uncertainty of ~ 28 %.
The results of the application of the algorithm described in the paragraphs above are presented
in the following section (Sect. 4) by means of maps, pie charts, plots and tables for each one of
the three basic regions of interest and their nine sub-regions.





## 4    Results and discussion
### 4.1    Validation of MODIS gridded data using ground-based observations
As discussed in Sects. 2 and 3, the high quality (QAC: 3) DT level-2 Collection 051 MODIS
data used in this work were validated in detail against data from 13 AERONET stations (see
Fig. 1). The stations were selected to make sure that their version 2.0 level 2.0 high quality
cloud  screened Cimel sunphotometric observations were covering at least 2 years and there
were at least 100 common days of AERONET and MODIS observations. The exact
geolocation of the AERONET stations is given in Table 1 (see also Fig.1) along with the
period of available data, the hosting country, the type of the station (e.g. urban/rural,
coastal/continental, etc.) and the corresponding mean overpass time of Terra and Aqua
MODIS. First, we collocated spatially and temporally the MODIS and AERONET
observations by temporally averaging AERONET measurements within ±30 min from the
MODIS overpass time (see Levy et al., 2010) and spatially averaging MODIS measurements
centered within a 25 x 25 km$^2$ window around each station (see Koukouli et al., 2010). The
use of a collocation window equal to the one used for the gridding procedure, practically,
allows us to validate at the same time the 0.1$^o$ x 0.1$^o$ MODIS gridded product.
The regression lines between MODIS and AERONET AODs are shown in Fig. 3 while
details about the validation results can be found in Table 2. Overall, the MODIS Terra DT
Collection 051 data overestimate $AOD_{550}$ by 11.59 % (Normalized Mean Bias - NMB) with
63.28 % of the data falling within the expected error (EE) envelope and 67.78% within the
pre-launch expected error (plEE) envelope. The expected error envelope is define as: AOD -
$|EE| \leq AOD_{MODIS} \leq AOD + |EE|$ with EE being ±(0.05±0.15AOD) (Levy et al., 2010) and
plEE being ±(0.05±0.20AOD) (Kaufman et al., 1997). On the other hand, the MODIS Aqua
DT Collection 051 data overestimate $AOD_{550}$ by 25.18 % (NMB) with 57.14 % of the data
falling within the EE envelope and 61.87 % within the plEE envelope. The percentage of the
MODIS Terra and Aqua data falling within the EE envelope are close to the 57 % given in
Remer et al. (2005) for Eastern Mediterranean. The validation results for each station
separately can be found in Table S1 of the Supplement. The results discussed in this
paragraph are comparable to the ones appearing in previous studies focusing on the
Mediterranean region (see Papadimas et al., 2009; Koukouli et al., 2010). In general, it is
shown here that the MODIS Terra Collection 051 data exhibit a better agreement with the
ground-based observations from AERONET than MODIS Aqua data do.



To be in line with the global validation of the DT Collection 051 product by Levy et al.
(2010) we also performed a validation with the specifications used in their work. We used a
50 x 50 km$^2$ window for the spatial collocation of the MODIS and AERONET data while
only days with at least 5 MODIS retrievals and 2 AERONET measurements were taken into
account. The increased size of the collocation window improves the results of the validation.
As shown in Table 2, MODIS Terra DT Collection 051 data overestimate $AOD_{550}$ by 5.10 %
(NMB) with 70.17 % of the data falling within the EE envelope and 74.64 % within the plEE
envelope. For MODIS Aqua, the NMB is 15.34%, while the percentage of the measurements
falling within the EE and plEE envelope is 66.76 % and 70.45 %, respectively. These results
about Eastern Mediterranean are close to the global ones presented in Levy et al. (2010).
As discussed in Sect. 3.1, data from the DB algorithm were used over bright arid and semi-
arid regions of North Africa for the production of the 0.1$^o$ x 0.1$^o$ MODIS gridded dataset for
grid cells with no DT data. Therefore, in this work we also perform a validation of the DB
Collection 051 product over the region of Eastern Mediterranean. In the case of DB data, we
first make use of all the available DB observations without any quality filtering over the 13
AERONET stations. A spatial window of 25-30 km has been typically used in the past for the
collocation of MODIS DB data with the AERONET observations (see Shi et al., 2011;
Ginoux et al.; 2012; Sayer et al., 2013; 2014) which is in line with the 0.25 x 0.25 km$^2$
window used here. The MODIS Terra DB data overestimate $AOD_{550}$ by 21.38 % (NMB) with
51.90 % of the data falling within the expected uncertainty envelope assuming a DB expected
uncertainty of $\pm 0.05 \pm 20\% AOD_{AERONET}$ (Hsu et al., 2006). The MODIS Aqua DB Collection
051 data overestimate $AOD_{550}$ by 33.03 % (NMB) with 55.30 % of the data falling within the
expected uncertainty envelope. We repeated the validation procedure for DB data taking into
account the highest quality data only. The sample of available measurements was diminished
by a factor of 5 in the case of MODIS Terra and 6 in the case of MODIS Aqua but the results
were pretty similar with the ones for the unfiltered data. Therefore, the use of unfiltered DB
data during the gridding procedure does not insert any significant uncertainty. The DB results
for the 13 AERONET stations examined here are not of the same agreement than the DT
results and the ones presented in previous studies utilizing DB Collection 051 data for other
stations and larger regions (see Shi et al., 2011; Ginoux et al., 2012). However, it has been
reported that stations in the region (e.g. Sede Boqer in Israel) are among the ones with the
greatest discrepancies between MODIS DB and AERONET measurements (Ginoux et al.,
2012). Nevertheless, as commented in Sect. 3.1, the DB data constitute only a small fraction





of the data used for the production of the MODIS gridded dataset (~ 1 % only of the 30000
grid cells covering Eastern Mediterranean has only DB retrievals) and therefore they do not
affect significantly its quality.
As discussed in Sect. 3.1 the gridding procedure was repeated four times using a gridding
window of 25, 50, 75 and 100-km using MODIS Terra $AOD_{550}$ data for the year 2004 showing
that the 25-km window is optimal for capturing local pollution sources. In order to see how the
size of the gridding window affects the agreement between MODIS and AERONET data we
also proceeded to a validation of MODIS DT data against AERONET measurements using
different spatial collocation windows (25, 50, 75 and 100-km) and two quality criteria, a "strict"
one: at least 2 AERONET measurements for each MODIS-AERONET pair and "stricter" one:
at least 5 MODIS retrievals and 2 AERONET measurements for each MODIS-AERONET pair
as in Levy et al. (2010). The results for the DT MODIS Terra and Aqua data are presented in
Table S2 of the Supplement. In general, it is shown that the increased size of the spatial
collocation window leads to an improvement of the bias between satellite and ground-based
observations. This is probably due to the inclusion of more observations into the calculations
which diminishes the noise of the MODIS observations. In addition, as expected, the stricter
quality criteria lead to a better agreement between MODIS DT and AERONET data. Taking
into account not only the NMB but also the regression lines and the other metrics appearing in
Table 2S, it is concluded that the 50-km window is the best choice for the validation procedure
in line with Ichoku et al. (2002). On the other hand, the 25-km validation results are close to the
50-km ones (see Table S2) and at the same time the 25-km gridding window allows for a more
efficient detection of local aerosol sources as shown in Sect. 3.1. Taking into account this, we
suggest that the 25-km window used for the production of the 0.1° x 0.1° gridded MODIS
dataset is the optimal selection for studying the spatial variability of aerosols, preserving at the
same time the representativeness of the real aerosol load over each specific spot.
**4.2   Aerosol spatial variability and hot spots**
The $AOD_{550}$ spatial variability over the greater Eastern Mediterranean region for the period
2000-2012 as seen from the Terra MODIS 0.1° x 0.1° dataset is presented in Fig. 4. Several
aerosol hot spots that coincide with megacities (e.g. Cairo, Istanbul), large cities (e.g. Athens,
Ankara, Alexandria, Izmir, Thessaloniki) or even medium sized cities (e.g. Larissa,
Limassol), industrial zones (e.g. OSTIM Industrial Zone in Ankara, Turkey), power plant
complexes (e.g. Maritsa Iztok complex at the Stara Zagora Province in Bulgaria, Ptolemaida-





Kozani power plants in Western Macedonia, Greece), river basins (e.g. Evros river Basin at
the borders between Greece and Turkey), etc, can be detected on the map. Indicatively, in Fig
4 we give a list of 35 local particle pollution sources in the region; however, careful
inspection of this map and the seasonal maps presented in Fig. 6 allows for the detection of
many more aerosol sources. The results from the analysis of Aqua MODIS data are pretty
similar as shown in Fig. S10 of the Supplement. A significant number of the local aerosol
sources can also be detected on the OMI 2004-2012 tropospheric $NO_2$ and PBL $SO_2$ maps
given in Figs. 5a and b which reveals the origin of aerosols over these regions (e.g. traffic,
industrial activities, etc). However, there are regions of high aerosol load which cannot be
seen in Fig. 5a and b and vice versa which is indicative of the significant role of other
anthropogenic or natural processes that contribute to the local aerosol load (e.g. fires, soil dust
from agricultural activities or arid regions, Sahara dust transport).
The topography (Fig. 5c) and precipitation (see Fig. 5d for annual precipitation levels for the
period 2000-2012 from TRMM) are also major determinants of the local $AOD_{550}$ levels. For
example, regions with mountain ranges on the Balkan Peninsula (e.g. Pindus mountain range
in Greece, Dinaric Alps that run through Albania and the former Yugoslav republics, the
Balkan mountain range in Central Bulgaria) are characterized by low AODs (see Fig. 4). On
the contrary, regions of low altitude are generally characterized by higher AODs because the
majority of anthropogenic activities is usually concentrated there. Also, low altitude regions
surrounded by high mountains are characterized by higher AODs as aerosols cannot be easily
transported by the wind (e.g. the industrialized regions in Central Bulgaria which are confined
between the high Balkan and Rodopi mountain ranges). As precipitation is the major removal
mechanism of pollutants in the atmosphere, regions with high $AOD_{550}$ are in many cases
connected to low precipitation levels and vice versa (see Figs. 4 and 5d). A striking example
is the one of Anatolia in Central Turkey which is characterized by lower precipitation levels
and higher aerosol loads compared to the surrounding regions. Also, the low precipitation
levels are partly responsible for the high aerosol loads appearing over Northern Africa.
Overall, the mean $AOD_{550}$ for the whole period of interest is estimated at $0.215 \pm 0.187$ for
Terra and $0.217 \pm 0.199$ for Aqua MODIS for the Eastern Mediterranean region which is ~ 45
% higher than the global average appearing in recent studies (e.g. Kourtidis et al., 2015). Over
land higher mean AODs are generally recorded ($0.219 \pm 0.165$ for Terra and $0.239 \pm 0.189$
for Aqua MODIS) than over the ocean ($0.213 \pm 0.201$ for Terra and $0.202 \pm 0.205$ for Aqua





MODIS). All these values along with the mean AODs for the 9 sub-regions of interest
covering Eastern Mediterranean can be found in Table 3.
The $AOD_{550}$ spatial variability on a seasonal basis from MODIS Terra and Aqua is presented
in Fig. 6 along with the difference between the two products. The majority of the local aerosol
sources over land are more prominent in summer due to the limited washout by precipitation
(see also Papadimas et al., 2008) and also due to the enhanced photochemical production of
secondary organic aerosols (Kanakidou et al., 2011 and references therein). In addition,
during summer, over the region, there is typically a significant transport of aerosols (e.g. see
Kanakidou et al., 2011 and references therein) and gaseous pollutants like $SO_2$ and $NO_2$ (see
Georgoulias et al., 2009; Zyrichidou et al., 2009) and biomass burning aerosols from Central-
Eastern Europe. Over the ocean, a profound maximum is observed in spring which is due to
the well documented transport of significant amounts of dust from the Sahara Desert
extending across the North African coast and the ocean. The seasonal variability of aerosols
and the relative role of different aerosol types and various processes is discussed in more
details in Sect 4.4.
The difference between MODIS Terra and Aqua Collection 051 $AOD_{550}$ over Eastern
Mediterranean is -0.002 (-1.40 %) for winter, -0.009 (-3.27 %) for spring, -0.011 (-4.46 %)
for summer and 0.008 (4.40 %) for autumn. $AOD_{550}$ levels from Terra MODIS are lower than
that from Aqua MODIS over land for all seasons. Over the sea, Terra MODIS $AOD_{550}$ levels
are lower than that of Aqua MODIS only in winter. The fact that Terra MODIS measurements
are systematically higher than that from Aqua over the ocean by ~ 0.01 on an annual basis is
in line with the findings of previous global studies for Collection 5 (e.g. Remer et al., 2006;
2008). Locally, one can see regions with positive and negative differences between Terra and
Aqua MODIS $AOD_{550}$. The patterns of the Terra-Aqua difference per season are presented in
Figs. 6c, f, i and l while the patterns of the percent difference are given in Fig. S11 of the
Supplement. The largest part of the Terra-Aqua MODIS differences over land and ocean
observed here may be attributed to the known calibration and sensor degradation issues of
MODIS (for details see Levy et al., 2010; 2013; Lyapustin et al., 2014). A significant effort
has been undertaken to address these issues in the new (Collection 6) MODIS product (e.g.
Levy et al., 2013; Lyapustin et al., 2014) and a repetition of a similar analysis with Collection
6 data in the future would be a valuable contribution. Taking into account the aforementioned
issues and the retrieval uncertainty of MODIS it becomes more than obvious that the
attribution of observed differences between Terra and Aqua to the diurnal variability of



aerosol load (e.g. over biomass burning regions) in the region is a difficult task. It is shown in
Fig. S12 of the Supplement that the diurnal variability of $AOD_{550}$ from AERONET ranges
significantly from station to station. The average hourly departure from the daily mean for the
total of the 13 stations ranges from ~ -5 % to ~ 5 %. Specifically, for the MODIS Terra and
Aqua overpass times, the AERONET $AOD_{550}$ difference ranges from ~ -10 % to ~ 10 % (see
Fig. S12b). The Terra-Aqua $AOD_{550}$ difference is negative for the total of the 13 stations
ranging from ~ -25 % to ~ -5 %. It is shown in Fig. S12b that the two differences exhibit a
similar variability from station to station which indicates that part of the observed Terra-Aqua
difference is indeed due to the diurnal variability of aerosols. However, as mentioned above,
the diurnal variability of aerosols is a very delicate issue and should be comprehensively
addressed in a future study. The same stands for other kind of variabilities which could be
connected to local and regional anthropogenic activities like e.g. the weekly cycle of aerosols
(see Georgoulias and Kourtidis, 2011; Georgoulias et al., 2015).
**4.3   Contribution of different aerosol types to the total $AOD_{550}$**
**4.3.1   Annual contribution**
As mentioned above, we attempt to estimate in our work the contribution of different aerosol
types to the total $AOD_{550}$ over the region of Eastern Mediterranean was calculated following
the methodology presented in Sect. 3.2. For the land covered areas, based on MODIS Terra
observations, we estimate that 52 % (0.112±0.087) of the total $AOD_{550}$ is due to
anthropogenic aerosols, 32 % (0.074±0.080) due to dust and 16 % (0.034±0.026) due to fine
mode natural aerosols (see Fig. 7). For the oceanic areas, 41 % (0.086±0.085) of the total
$AOD_{550}$ is due to anthropogenic aerosols, 34 % (0.076±0.185) due to dust and 25 %
(0.054±0.018) due to marine aerosols (see Fig. 7). The results based on observations from
MODIS Aqua are similar. Over land, 50 % (0.117±0.093) of the total $AOD_{550}$ is
anthropogenic, 35 % (0.090±0.102) is due to dust and 15 % (0.035±0.028) due to fine mode
natural aerosols, while, over ocean, 40 % (0.079±0.080) of the total $AOD_{550}$ is of
anthropogenic origin, 33 % (0.070±0.181) is due to dust and 27 % (0.054±0.018) due to
marine aerosols (see Fig. 7). These results along with the relative contributions and the annual
$\tau_a$, $\tau_d$, $\tau_n$ and $\tau_m$ levels for each one of the nine sub-regions of interest (see Fig. 1) are given in
Table 4.
For anthropogenic aerosols, the region with the highest relative contribution is NBL (59 % for
both Terra and Aqua MODIS) while the region with the lowest relative contribution is SWO



(32 % for both Terra and Aqua MODIS) (see also Table 4). The spatial variability of $\tau_\alpha$ is
presented in Fig. 8a for Terra MODIS and Fig. S13a of the Supplement for Aqua MODIS, the
patterns being similar in both cases. Over land, the annual $\tau_a$ patterns are similar to the
AOD$_{550}$ patterns, the highest values appearing over local particle pollution sources (cities,
industrial zones, etc.). Over the sea, $\tau_a$ is higher along the coasts, while it drops significantly
towards other directions. An interesting feature here is that the oceanic region of Black Sea
(BSO) presents higher relative anthropogenic contributions than the rest of the oceanic sub-
regions but also than land areas with significant anthropogenic sources (e.g. ANL and NAL).
This is indicative of the transport of atmospheric particles from Central Europe and biomass
burning aerosols during the biomass burning seasons in April-May from Russia (across the
latitudinal zone 45$^o$N-55$^o$N) and July-August from South-Western Russia and Eastern Europe
(Amiridis et al., 2010). These aerosols are transported at much lower latitudes as shown in
previous studies (e.g. Vrekoussis et al., 2005; Karnieli et al., 2009) reaching the Sahara Desert
and the Middle East regions (Pozzer et al., 2015). The fact that $\tau_a$ drops gradually from the
coasts is also seen in Fig. 9 where the latitudinal variability of the optical depths of the
different aerosol types ($\tau_a$, $\tau_d$, $\tau_n$ and $\tau_m$) is presented for four bands that cover the whole
Eastern Mediterranean. An interesting feature is that $\tau_a$ increases nearby the shoreline
(particularly along the North African coastal zone) before it gradually decreases. Over land
aerosols are located within the atmospheric boundary layer, close to the emission sources, and
hence, their deposition and removal from the atmosphere is more efficient than over the
ocean. The particles which are transported over the ocean on the other hand usually reach
greater heights which prolongs their lifetime.
As shown in Fig. 9, the same feature is observed for dust. Indicatively, $\tau_d$ and the relative
contribution of dust to the total AOD$_{550}$ on an annual basis over the oceanic regions of SWO
and SEO are in general higher or comparable to the ones over NAL (see Table 4 for more
details). In Fig. 9, the MODIS-based $\tau_d$ latitudinal variability is presented along with the
latitudinal variability of dust AOD$_{532}$ and extinction coefficients of dust at 532 nm from
LIVAS. As expected, in all cases $\tau_d$ decreases with distance from the large dust sources in the
South and South-East (Sahara Desert, Middle East deserts) with local maxima over the
latitudinal zone from 35$^o$N to 40$^o$N (especially for band 2  and band 3). The latitudinal
variability of $\tau_d$ is similar to the latitudinal variability of dust AOD$_{532}$ for all the four bands
despite the fact that the MODIS-based data have a resolution 100 times higher (0.1$^o$ vs 1$^o$) and
therefore are more sensitive to local characteristics. Dust reaches heights up to ~ 4-5 km in the





area; however, the largest fraction of dust mass is confined within the first 2-3 km of the
troposphere (see Fig. 9). The annual $\tau_d$ patterns are shown in Fig. 8b for Terra MODIS (Fig.
S13b of the Supplement for Aqua MODIS). The main dust transport pathways over the
oceanic areas of Eastern Mediterranean can be seen along with various local maxima over
land. The highest $\tau_d$ values over land appear over the regions of NAL and ANL (see Table 4)
and along the coasts. The high dust concentrations appearing over these regions are not only
due to the transport of dust from the nearby deserts but also due to local dust sources. A
recent study by Liora et al. (2015) reports various local sources of wind blown dust along the
coastal regions of Greece and Turkey, over the region of Anatolia in Turkey, over the Greek
islands, Crete, Cyprus and regions close to the coastal zone of Middle East. Their results are
in good agreement with the $\tau_d$ patterns presented in this work.
As shown in Fig. 7, fine mode natural aerosols exhibit the lowest contribution to the total
$AOD_{550}$ compared to the other aerosol types over land. The spatial variability of $\tau_n$ is very low
compared to $\tau_a$ and $\tau_d$ as shown in Figs. 8c and 9. It is inferred from the values appearing in
Table 4 that $\tau_n$ increases slightly as one moves from North to South; however, the relative
contribution of fine mode natural aerosols to the total $AOD_{550}$ slightly decreases (i.e. 17.67 %
over NBL and 14.97 % over NAL according to Terra MODIS observations). The latitudinal
variability and the percentages appearing in Table 4 are in accordance to the relative
contributions of biogenic aerosols to the total $AOD_{550}$ appearing over Eastern Mediterranean
in a recent modeling study (Rea et., 2015).
Similar to fine mode natural aerosols over land, marine aerosols generally have the lowest
contribution to the total $AOD_{550}$ compared to the other aerosol types over the sea (see Fig. 7
and Table 4) except for BSO. The variability of $\tau_m$ is very low compared to $\tau_a$ and $\tau_d$. On an
annual basis, high $\tau_m$ values appear over the Aegean Sea and the oceanic area between Crete
and the North African coast while slightly lower values appear along the coasts of Eastern
Mediterranean (see Figs. 8d and 9). The $\tau_m$ patterns follow the near surface wind speed
patterns in the region (see Fig. S14 of the Supplement) being in accordance to the $\tau_m$, marine
particulate matter concentration or sea salt emission patterns appearing in other studies (Im et
al., 2012; Nabat et al., 2013; Rea et al., 2015; Liora et al., 2015).
**4.3.2  Seasonal contribution**
The contribution of different aerosol types to the total $AOD_{550}$ over the Eastern Mediterranean
varies from season to season. The relative contribution of each aerosol type over EML and



EMO for each season is shown in Fig. 10. Over land, the relative contribution of $\tau_a$, $\tau_d$ and $\tau_n$
to the total $AOD_{550}$ exhibits a low seasonal variability. The relative contribution of
anthropogenic aerosols to the total $AOD_{550}$ ranges from 49 % in SON to 50 % in DJF based
on Terra MODIS observations and from 48 % in MAM and SON to 52 % in JJA based on
Aqua MODIS observations. In contrast, over the oceanic regions the relative contribution of
$\tau_a$, $\tau_d$ and $\tau_m$ to the total $AOD_{550}$ exhibits a significant seasonal variability. The relative
contribution of anthropogenic aerosols to the total $AOD_{550}$ ranges from 27 % / 27 % in DJF to
50 % / 47 % in JJA based on Terra/Aqua MODIS observations. The percentages appearing
here are in accordance to the values appearing in Hatzianastassiou et al. (2009) where a
different satellite-based approach was followed. Indicatively, for the greater Athens area, an
average summertime anthropogenic contribution of ~ 50 % was found here based on Terra
MODIS data which is within the summer period range of 47-61 % indicated in the study by
Hatzianastassiou et al. (2009). In addition, the corresponding values for the greater
Thessaloniki area, Crete, Cairo and Alexandria are 53 %, 38 %, 48 % and 41 %, respectively,
within the range of values (57-73 %, 36-52 %, 34-56 % and 23-60 %) shown in
Hatzianastassiou et al. (2009). Only in the case of Ankara, our results suggest a lower
anthropogenic contribution (52 % versus 71-84 %). Particularly for Athens, Gerasopoulos et
al. (2011) following a different approach incorporating ground-based AOD observations and
trajectory modeling reached similar results (annual contribution of ~ 62 % from local and
regional sources and continental Europe which is expected to be mostly of anthropogenic
origin). Similarly, for Crete, Bergamo et al. (2008) using a different approach, also utilizing
ground-based data, found an annual anthropogenic contribution of ~ 43 %.
The seasonal patterns of the anthropogenic aerosols ($\tau_a$) over the Eastern Mediterranean based
on MODIS Terra observations are presented in Figs. 11a, e, i and m while the seasonal
variability of $\tau_a$ over the whole region, over the land covered part and the oceanic part and
over the 9 sub-regions of interest is presented in Fig. 12. The results based on MODIS Aqua
observations are similar and can be found in Figs. S15a, e, i and m and Fig. S16 of the
Supplement. Generally, the local hot spots are detectable throughout the year; however, they
are becoming much more discernible in spring and especially in summer. As shown in Fig.
12a, $\tau_a$ nearly doubles during the warm period of the year (spring-summer) with the seasonal
variability being stronger over the sea (Fig. 12c) than over land (Fig. 12b). A clear peak is
observed in summer, August being the month with highest $\tau_a$ levels. As discussed in Sect.
4.3.1 the summer peak is mostly a result of three basic reasons. The first one is the deficiency



of wet removal processes compared to the cold period. As shown in Fig. S17, based on the
TRMM satellite observations, August and July are the months with the lowest precipitation
levels over the land covered part (a drop of ~ 75 % compared to winter months) and over the
oceanic part (a drop of ~ 90 % compared to winter months) of Eastern Mediterranean,
respectively. The second reason is the enhancement of the photochemical production of
secondary organic aerosols in summer (Kanakidou et al., 2011) and the third reason is the
transport of pollution aerosols from Central Europe and biomass burning aerosols from South-
Western Russia and Eastern Europe during the biomass burning season in July-August
(Amiridis et al., 2010). The Etesians, which are persistent northerly winds that prevail over
Eastern Mediterranean during summer, bring dry and cool air masses and aerosols from the
regions mentioned above while blocking at the same time the transport of desert dust in the
region and dispersing local pollution in urban areas to levels typical for rural areas (see Tyrlis
and Lelieveld, 2013 and references therein). As seen in Figs. 12a-l, a smaller but distinct in
most cases $\tau_a$ peak appears in April mostly as a result of the transport of biomass burning
aerosols from Russia (across the latitudinal zone $45^{o}N$-$55^{o}N$). This is in line with the findings
of Sciare et al. (2008) who detected traces of these biomass burning aerosols at the island of
Crete in Southern Greece.
As discussed above, the relative contribution of dust to the total $AOD_{550}$ over land exhibits a
low seasonal variability ranging from 29 % in DJF to 36 % in SON based on Terra MODIS
observations and from 33 % in JJA to 38 % in SON  based on Aqua MODIS observations (see
Fig. 10). Over the oceanic regions the relative contribution of dust to the total $AOD_{550}$ ranges
significantly throughout a year from 26 % / 28 % in JJA to 42 % / 39 % in MAM based on
Terra/Aqua MODIS observations. The percentages appearing here are in accordance to model
and observational studies. For example, de Meij et al. (2012) using the atmospheric chemistry
general circulation model EMAC (ECHAM/MESSy Atmospheric Chemistry) showed that
dust contributes on an annual level ~ 30 % to the total $AOD_{550}$ over stations located in the
area of Eastern Mediterranean. Gerasopoulos et al. (2011) found a ~ 23 % percent
contribution of North African dust to the total AOD over Athens using ground-based AOD
observations and trajectory modeling. Taking into account that part of the ~ 39 % local and
regional sources appearing in Gerasopoulos et al. (2011) is due to local dust sources,
especially in summer, turns out that their results are in agreement with the ~ 33 % relative
contribution found in this work for the greater Athens area based on Terra MODIS
observations. The seasonal patterns of dust ($\tau_d$) over the region based on Terra MODIS





observations are shown in Figs. 11b, f, j and n while the seasonal variability of $\tau_d$ over the
whole region, over land, over the sea and over the 9 sub-regions of interest is shown in Fig.
12. The corresponding results based on MODIS Aqua observations are pretty similar and can
be found in Figs. S15b, f, j and n and Fig. S16 of the Supplement.
As seen in Fig. 11f, in spring, mostly due to the strong Sahara dust events, very high $\tau_d$ values
appear over land regions in North Africa, Middle East, Anatolia and oceanic areas across
Eastern Mediterranean (especially below $35^{o}$N). Dust loading over the sea exhibits two
maxima, one at the coastal zone of Libya and one across the coastal zone of Middle East. The
same two maxima but with much lower $\tau_d$ values appear in summer (Fig. 11j) and autumn
(Fig. 11n). Over land, the $\tau_d$ patterns are similar in summer and autumn, the maximum values
appearing over the Anatolian Plateau and areas of North Africa and Middle East. During
winter, dust maxima appear across the coastal zone of Northern Africa with relatively low $\tau_d$
values across the coastal zone of Middle East (Fig. 11b). In winter $\tau_d$ levels are low over land
compared to the other seasons (Figs. 11b, f, j and n) as precipitation levels (see Fig. S17 of the
Supplement) and hence wet scavenging of aerosols peak. At the same time, the local
emissions of dust are low for regions away from the large area sources in the South (Liora et
al., 2015). In contrast, over the sea $\tau_d$ levels in winter are similar or slightly higher for some
areas than that in summer and autumn (see Figs. 11 and 12) as this is the season with the
second highest frequency (after spring) of strong (~ 21 %) and extreme (~ 26 %) desert dust
episodes in the region (see Gkikas et al., 2013 for details). February is by far the winter month
the highest $\tau_d$ levels (see Fig. 12) in line with the findings of Pey et al. (2013) who showed
that the intensity of African dust episodes over stations in Greece and Cyprus peaks in
February. Dust exhibits a strong peak in spring, April being the month with the highest $\tau_d$
levels in line with other studies (e.g. Israelevic et al., 2012; Varga et al., 2014). The peak in
April is a result of the high cyclonic activity over North Africa during this month as shown by
Flaounas et al. (2015). According to the same study, low pressure systems are responsible for
~ 10-20 % of moderate and ~ 40-50 % of high and extreme Sahara dust transport events over
Eastern Mediterranean. North Africa (Sharav) cyclones develop mainly in spring and summer
while Mediterranean cyclones develop in winter and autumn. The Mediterranean cyclones are
more intense than Sharav cyclones. The region, is also affected by events bringing particles
from dust source regions in the eastern part of Mediterranean basin (Negev desert in Israel,
Sinai in Egypt, Anatolian Plateau in Turkey) and the Arabian deserts (Basart et al. 2009; Pey
et al., 2013; Abdelkader et al., 2015). Dust remains in the atmosphere for a period of 1-4 days



undergoing chemical aging before being removed (see Abdelkader et al. 2015 and references
therein). The seasonal variability of $\tau_d$ is much stronger and the spring maxima much more
prominent over the sea (see Fig. 12). This is expected, as dust is only occasionally transported
over the sea during episodic events, while over land, local sources also contribute to the dust
burden especially in summer due to the dryness of soil. For example, over NBL, a broad
spring-summer peak is observed, June being the month with the highest $\tau_d$ levels. As one
moves south (SBL, ANL and NAL) the April peak becomes more prominent.
The relative contribution of fine mode natural aerosols to the total $AOD_{550}$ over land exhibits
a very low seasonal variability ranging from 15 % in MAM and SON to 16 % in DJF and JJA
based on Terra MODIS observations and from 14 % in DJF and SON to 15 % in MAM and
JJA based on Aqua MODIS observations (see Fig. 10). The seasonal variability is also very
low, the highest values appearing in spring and summer (Fig. 12). Despite the generally low
contribution of fine mode natural aerosols to the total $AOD_{550}$ over Eastern Mediterranean, $\tau_n$
levels are similar to $\tau_d$ levels during winter months over specific regions (NBL and SBL). The
low seasonal variability can also be seen in Figs. 11c, g, k and o where the patterns of fine
mode natural aerosols ($\tau_n$) are presented.
The seasonal relative contribution of marine aerosols to the total $AOD_{550}$ over the oceanic
regions of Eastern Mediterranean is shown in Fig. 10. $\tau_m$ ranges from 20 % in MAM to 35 %
in DJF based on Terra MODIS observations and from 21 % in MAM to 36 % in DJF based on
Aqua MODIS observations (see Fig. 10). Like in the case of fine mode natural aerosols, the
seasonal variability is very low, but here the highest values appear in winter (Fig. 12). Due to
the linear relation of $\tau_m$ and near surface wind speed within our algorithm (see Fig. 2) the $\tau_m$
seasonal variability and patterns follow the wind speed ones (see Figs. 11d, h, l, p and S14).
Marine aerosol concentrations are lower close to the coastlines while the highest
concentrations (see Liora et al., 2015) and $\tau_m$ values within Eastern Mediterranean appear
over the Aegean Sea (see Fig. 11). Overall, the $\tau_m$ patterns are in accordance to the $\tau_m$, marine
particulate matter concentration and sea salt emission patterns from previous studies (Im et
al., 2012; Nabat et al., 2013; Rea et al., 2015; Liora et al., 2015).
**5   Conclusions**
In this work, satellite data from MODIS Terra (3/2000-12/2012) and Aqua (7/2002-12/2012)
were analyzed separately in order to examine the spatial and temporal variability of aerosols
over the region of Eastern Mediterranean. A high resolution (0.1$^{\circ}$ x 0.1$^{\circ}$) MODIS gridded





dataset was compiled using a method that could be used in future regional studies. A number
of tests were implemented and the dataset was validated in detail using sunphotometric
observations from 13 AERONET stations. It is shown that the gridding method we use offers
the best compromise for studying the spatial variability of aerosols on a regional or local
scale, preserving at the same time the representativeness of the real aerosol load over each
specific spot.
Based on MODIS observations the average $AOD_{550}$ levels over the region of Eastern
Mediterranean are ~ 0.22 ± 0.19 which is ~ 45 % higher than the global mean. A number of
aerosol hot spots that coincide with megacities, large and even medium size cities, industrial
zones, power plant complexes, river basins, etc., can be detected on the AOD maps. A
number of local aerosol sources can also be seen on satellite retrieved tropospheric $NO_2$ and
planetary boundary layer $SO_2$ maps from OMI/AURA. This is indicative of the strong
presence of anthropogenic aerosols over these regions. Topography and precipitation also
play an important role. Generally, regions with mountain ranges are characterized by low
AODs while regions of low altitude are characterized by higher AODs. Regions with high
$AOD_{550}$ are in many cases connected to low precipitation levels and vice versa as is the major
washout mechanism of atmospheric pollutants.
The $AOD_{550}$ patterns over Eastern Mediterranean exhibit a significant seasonal variability
which is mostly driven by precipitation, photochemical production of secondary organic
aerosols, transport of pollution and biomass burning aerosols from Central and Eastern
Europe and transport of dust from the Sahara Desert and the Middle East. Differences
between MODIS Terra and Aqua Collection 051 $AOD_{550}$ over the Eastern Mediterranean are
generally small (~ -8 % over land and ~ 5 % over the sea). The comparison of the Terra-Aqua
differences with diurnal variabilities from the AERONET stations showed that only a part of
the observed differences is due to the diurnal variability of aerosols.
The MODIS data were combined with data from other satellites (Earth Probe TOMS,
OMI/AURA), reanalysis projects (ERA-Interim, MACC) and a chemistry-aerosol-transport
model (GOCART) to calculate the contribution of different types of aerosols to the total
$AOD_{550}$. The algorithm used was optimized for the Eastern Mediterranean through a number
of tests and comparison with LIVAS CALIOP/CALIPSO dust retrievals and AERONET
ground-based observations. A different approach is used for land and ocean covered areas as
there is not any reliable satellite retrieved quantity to separate the contribution of fine and
coarse mode aerosols over water surfaces.





Overall, for the land areas, based on MODIS Terra observations, 52 % (0.112±0.087) of the
total $AOD_{550}$ is due to anthropogenic aerosols, 32 % (0.074±0.080) due to dust and 16 %
(0.034±0.026) due to fine mode natural aerosols (see Fig. 7). For the oceanic areas, 41 %
(0.086±0.085) of the total $AOD_{550}$ is due to anthropogenic aerosols, 34 % (0.076±0.185) due
to dust and 25 % (0.054±0.018) due to marine aerosols. The results based on observations
from MODIS Aqua are in accord with previous studies.
Over land, the $\tau_a$ maxima are detected over local particle pollution sources (cities, industrial
zones, etc.). Over the sea, $\tau_a$ is higher along the coasts being significantly lower at greater
distance. Very high $\tau_d$ values appear over land regions in North Africa, Middle East, Anatolia
and oceanic areas across Eastern Mediterranean, especially for latitudes below $35^{o}N$. Over the
sea, dust loading exhibits two maxima, one at the coastal zone of Libya and one across the
coastal zone of the Middle East. $\tau_d$ decreases with distance from the large dust sources in the
South and South-East. Generally, dust reaches heights up to ~ 4-5 km in the area, the largest
fraction of dust mass being confined within the first 2-3 km of the troposphere. The spatial
variability of $\tau_n$ and $\tau_m$ is very low compared to $\tau_a$ and $\tau_d$, following the total $AOD_{550}$ patterns
and the near surface wind speed patterns, respectively.
Over land, the relative contribution of anthropogenic aerosols, dust and fine mode natural
aerosols to the total $AOD_{550}$ exhibits a low seasonal variability, while over the sea the relative
contribution of anthropogenic aerosols, dust and marine aerosols shows a significant seasonal
variability.
$\tau_a$ nearly doubles during the warm period of the year (spring-summer), August and April
being the months with the highest $\tau_a$ levels. The summer peak is mostly the result of low
precipitation levels, enhancement of the photochemical production of secondary organic
aerosols and transport of pollution aerosols from Central Europe and biomass burning
aerosols from South-Western Russia and Eastern Europe during the biomass burning season
in July-August. The spring maximum in April is mostly the result of transport of biomass
burning aerosols from Russia in line with previous studies. Dust exhibits a strong peak in
spring (April), especially over the southern regions. April is the month with the highest $\tau_d$
levels as a result of the high cyclonic activity over North Africa. The seasonal variability of
dust is much stronger and the spring maxima much more prominent over the sea as dust is
only occasionally transported there during episodic events, while over land, local sources
contribute to the dust burden, especially in summer due to the soil dryness. The seasonal
variability of fine mode natural aerosols is very low, the highest values appearing in spring



and summer. Marine aerosols also present a very low seasonal variability, the highest values
appearing in winter due to the high near surface wind speeds.
Overall, it is suggested that the $AOD_{550}$, $\tau_a$, $\tau_d$, $\tau_n$ and $\tau_m$ high resolution gridded dataset which
was compiled in this work could be used in a number of future atmospheric and biological
studies focusing on the region of Eastern Mediterranean (e.g. satellite and ground-based
studies on aerosol-cloud-radiation interactions, experimental and field campaign studies on
aerosols and clouds and research on the impact of aerosols on human health and nature). It is
also acknowledged that a future update of the results presented here using more recent
releases of MODIS aerosol data (e.g. Collection 6) and aerosol reanalysis datasets (e.g.
NASA's Modern-Era Retrospective Analysis For Research And Applications Aerosol Re-
analysis) would be a useful contribution.
**Acknowledgements**
This research received funding from the European Social Fund (ESF) and national resources
under the operational programme Education and Lifelong Learning (EdLL) within the
framework of the Action "Supporting Postdoctoral Researchers" (QUADIEEMS project),
from the European Research Council under the European Union's Seventh Framework
Programme (FP7/2007-2013)/ERC grant agreement no. 226144 (C8 project), from the FP7
Programme MarcoPolo (grant number 606953, theme SPA.2013.3.2-01) and from the
European Union's Horizon 2020 research and innovation programme under grant agreement
no. 654109. The authors express their gratitude to the teams that developed the algorithms and
produced the satellite products used in this study and to those who worked on the production
of the model and reanalysis data used here. Special thanks are expressed to NASA Goddard
Space Flight Center (GSFC) Level 1 and Atmosphere Archive and Distribution System
(LAADS) (http://ladsweb.nascom.nasa.gov) for making available the MODIS Terra and Aqua
Collection 051 level-2 aerosol data and the principal investigators and staff maintaining the
13 AERONET (http://aeronet.gsfc.nasa.gov) sites used in the present work. LIVAS has been
financed under the ESA-ESTEC project LIVAS (contract no. 4000104106/11/NL/FF/fk). We
thank the ICARE Data and Services Center (www.icare.univ-lille1.fr) for providing access to
NASA's CALIPSO data and acknowledge the use of NASA's CALIPSO data. Special thanks
are expressed to ECMWF (www.ecmwf.int) for the provision of the ERA-Interim and MACC
reanalysis data. NASA's GIOVANNI web database (http://giovanni.gsfc.nasa.gov/giovanni/)
is highly acknowledged for the provision of Aerosol Index data from Earth Probe TOMS and



OMI, aerosol data from the GOCART chemistry-aerosol-transport model (older version of GIOVANNI), tropospheric $NO_2$ and PBL $SO_2$ columnar data from OMI and precipitation data from 3B43 TRMM and Other Sources Monthly Rainfall Product. A.K.G. acknowledges the fruitful discussions with various colleagues from the Max Planck Institute for Chemistry and the Cyprus Institute (EEWRC) who indirectly contributed to this research.

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



**Table 1.** Full name, abbreviation, geolocation, host country and type of the 13 AERONET
cimel sunphotometer sites used for the validation of MODIS Terra and Aqua Collection 051
observations. The common measurement period of MODIS and AERONET data and the
corresponding overpass time of MODIS Terra and Aqua (Italics) over each station are also
given.

| AERONET Station | Lat (°N) | Lon (°E) | Period of study | Country | Type | TERRA overpass | *AQUA overpass* |
|---|---|---|---|---|---|---|---|
| ATHENS-NOA (ATH) | 37.988 | 23.775 | 05/2008-10/2012 | Greece | Urban (coastal) | 9:23±22min UT | *11:32±22min UT* |
| Bucharest Inoe (BUC) | 44.348 | 26.030 | 07/2007-09/2012 | Romania | Sub-urban (coastal) | 9:17±24min UT | *11:15±20min UT* |
| CUT-TEPAK (CUT) | 34.675 | 33.043 | 04/2010-12/2012 | Cyprus | Urban (coastal) | 8:43±25min UT | *10:55±25min UT* |
| Eforie (EFO) | 44.075 | 28.632 | 09/2009/12/2012 | Romania | Rural (coastal) | 9:09±21min UT | *11:04±21min UT* |
| FORTH Crete (FOR) | 35.333 | 25.282 | 01/2003-08/2011 | Greece | Rural (coastal) | 9:12±24min UT | *11:25±25min UT* |
| IMS-METU-ERDEMLI (IMS) | 36.565 | 34.255 | 01/2004-01/2012 | Turkey | Rural (coastal) | 8:39±23min UT | *10:48±22min UT* |
| Lecce University (LEC) | 40.335 | 18.111 | 03/2003-12/2012 | Italy | Sub-urban (coastal) | 9:44±25min UT | *11:49±25min UT* |
| Nes ziona (NES) | 31.922 | 34.789 | 02/2000-12/2012 | Israel | Sub-urban (coastal) | 8:38±24min UT | *10:44±25min UT* |
| SEDE BOKER (SED) | 30.855 | 34.782 | 01/2000-04/2012 | Israel | Rural (semi-arid) | 8:30±27min UT | *10:50±25min UT* |
| Sevastopol (SEV) | 44.616 | 33.517 | 05/2006-12/2012 | Ukr.-Crimea | Urban (coastal) | 8:51±21min UT | *10:40±21min UT* |
| Thessaloniki (THE) | 40.630 | 22.960 | 09/2005-12/2012 | Greece | Urban (coastal) | 9:28±25min UT | *11:32±22min UT* |
| TUBITAK UZAY Ankara (TUB) | 39.891 | 32.778 | 12/2009-04/2012 | Turkey | Urban (continental) | 8:48±26min UT | *10:56±24min UT* |
| Xanthi (XAN) | 41.147 | 24.919 | 01/2008-10/2010 | Greece | Rural (coastal) | 9:18±25min UT | *11:24±21min UT* |



**Table 2.** Results of the comparison of spatially (using a spatial window around each station)
and temporally (±30 min from the MODIS overpass time) collocated MODIS Terra and Aqua
(Italics) Collection 051 level-2 and AERONET sunphotometric (quadratically interpolated)
$AOD_{550}$ observations for the Eastern Mediterranean stations. The algorithms used for the
production of the validated MODIS data (DT and DB), the spatial window used for the spatial
collocation (25 x 25 $km^2$ or 50 x 50 $km^2$ window around each station) with the AERONET
data, the average MODIS and AERONET $AOD_{550}$ and the corresponding ±1σ values, the
mean difference between them, the normalized mean bias (NMB) and the corresponding root
mean squared (RMS) error, the percentage of the collocation points that fall within the
expected error (EE) envelope and the pre-launch expected error (plEE) envelope (Expected
Uncertainty - EU envelope for DB data), the correlation coefficient R, the slope a and the
intercept of the regression line and the number of the collocation points are given in the table.
L10 denotes the use of a collocation window of $50^o$ x $50^o$ as in Levy et al. (2010) while HQ
denotes the use of high quality data only.

| Alg. | Window | MODIS TERRA *MODIS AQUA* | AERONET | Mean Diff. | NMB % | RMS err. | in EE % | in pl EE % | R | a | b | Obs |
|---|---|---|---|---|---|---|---|---|---|---|---|---|
| DT | 25 km | 0.223±0.163 | 0.200±0.123 | 0.023±0.106 | 11.59 | 0.11 | 63.28 | 67.78 | 0.76 | 1.007 | 0.022 | 6697 |
| *DT* | *25 km* | *0.247±0.173* | *0.197±0.121* | *0.050±0.109* | *25.18* | *0.12* | *57.14* | *61.87* | *0.78* | *1.113* | *0.027* | *6283* |
| DT | 50 km (L10) | 0.204±0.152 | 0.194±0.124 | 0.010±0.085 | 5.10 | 0.09 | 70.17 | 74.64 | 0.83 | 1.016 | 0.007 | 6054 |
| *DT* | *50 km (L10)* | *0.224±0.155* | *0.194±0.125* | *0.030±0.088* | *15.34* | *0.09* | *66.76* | *70.45* | *0.82* | *1.018* | *0.026* | *5557* |
| DB | 25 km | 0.226±0.177 | 0.186±0.128 | 0.040±0.162 | 21.38 | 0.17 | - | 51.90 | 0.47 | 0.657 | 0.104 | 2580 |
| *DB* | *25 km* | *0.242±0.217* | *0.182±0.118* | *0.06±0.196* | *33.03* | *0.20* | *-* | *55.30* | *0.44* | *0.815* | *0.094* | *5345* |
| DB_HQ | 25 km | 0.229±0.158 | 0.186±0.132 | 0.043±0.141 | 22.82 | 0.15 | - | 52.41 | 0.54 | 0.651 | 0.108 | 498 |
| *DB_HQ* | *25 km* | *0.260±0.220* | *0.186±0.138* | *0.074±0.204* | *39.84* | *0.22* | *-* | *52.34* | *0.42* | *0.670* | *0.136* | *896* |





**Table 3.** AOD$_{550}$ levels, the corresponding ±1σ values and the number of gridded values used
for the calculations over Eastern Mediterranean (EMT), over the land covered part (EML),
over the oceanic part and over the 9 sub-regions of Eastern Mediterranean appearing in Fig. 1
based on the MODIS Terra and Aqua (Italics) observations.

| Region | MODIS TERRA AOD$_{550}$ | Num. of values | MODIS AQUA AOD$_{550}$ | Num. of values |
|---|---|---|---|---|
| EMT | 0.215±0.187 | 61496654 | *0.217±0.199* | *49522934* |
| EML | 0.219±0.165 | 25923766 | *0.239±0.189* | *21008713* |
| EMO | 0.213±0.201 | 35572888 | *0.202±0.205* | *28514221* |
| NBL | 0.183±0.163 | 5563495 | *0.187±0.162* | *3853688* |
| SBL | 0.197±0.152 | 7345829 | *0.207±0.152* | *5272449* |
| ANL | 0.223±0.146 | 7948817 | *0.228±0.148* | *5539261* |
| NAL | 0.282±0.192 | 5065625 | *0.306±0.238* | *6343315* |
| BSO | 0.198±0.150 | 6433951 | *0.183±0.134* | *5262438* |
| NWO | 0.209±0.162 | 11645069 | *0.197±0.154* | *9231630* |
| SWO | 0.226±0.266 | 6202893 | *0.223±0.310* | *4925665* |
| NEO | 0.214±0.196 | 4807910 | *0.199±0.166* | *3896554* |
| SEO | 0.221±0.236 | 6483065 | *0.210±0.239* | *5197934* |



**Table 4.** Relative contribution of anthropogenic aerosols, dust, fine mode natural and marine
aerosols to the total $AOD_{550}$ (bold) and the corresponding τa, τd, τn, τm levels with their ±1σ
values (in parentheses) over Eastern Mediterranean (EMT), over the land covered part (EML),
over the oceanic part and over the 9 sub-regions of Eastern Mediterranean appearing in Fig. 1
based on the MODIS Terra and Aqua (Italics) observations. The sum of the aerosol type
AODs per region does not necessarily correspond to the total $AOD_{550}$ values appearing in
Table 3 as these results were for the total of the days with aerosol retrievals even for days
when our aerosol type separation algorithm was not applicable.

| Region | Satellite | Contribution to MODIS TERRA/*AQUA* $AOD_{550}$ | | | |
|---|---|---|---|---|---|
| | | Anthropogenic | Dust | Fine mode natural | Marine |
| EML | TERRA | **52 %** (0.112±0.087) | **32 %** (0.074±0.080) | **16 %** (0.034±0.026) | - |
| | *AQUA* | *50 % (0.117±0.093)* | *35 % (0.090±0.102)* | *15 % (0.035±0.028)* | *-* |
| EMO | TERRA | **41 %** (0.086±0.085) | **34 %** (0.076±0.185) | - | **25 %** (0.054±0.018) |
| | *AQUA* | *40 % (0.079±0.080)* | *33 % (0.070±0.181)* | *-* | *27 % (0.054±0.018)* |
| NBL | TERRA | **59 %** (0.108±0.101) | **23 %** (0.042±0.046) | **18 %** (0.032±0.030) | - |
| | *AQUA* | *59 % (0.110±0.100)* | *24 % (0.045±0.047)* | *17 % (0.033±0.030)* | *-* |
| SBL | TERRA | **55 %** (0.109±0.088) | **28 %** (0.056±0.058) | **17 %** (0.033±0.026) | - |
| | *AQUA* | *55 % (0.113±0.088)* | *29 % (0.060±0.060)* | *16 % (0.034±0.026)* | *-* |
| ANL | TERRA | **51 %** (0.113±0.075) | **34 %** (0.076±0.068) | **15 %** (0.034±0.023) | - |
| | *AQUA* | *50 % (0.114±0.075)* | *35 % (0.079±0.070)* | *15 % (0.034±0.023)* | *-* |
| NAL | TERRA | **50 %** (0.113±0.083) | **35 %** (0.083±0.085) | **15 %** (0.034±0.025) | - |
| | *AQUA* | *48 % (0.118±0.091)* | *38 % (0.099±0.108)* | *14 % (0.035±0.027)* | *-* |
| BSO | TERRA | **53 %** (0.108±0.103) | **22 %** (0.044±0.101) | - | **25 %** (0.051±0.016) |
| | *AQUA* | *51 % (0.094±0.087)* | *22 % (0.042±0.085)* | *-* | *27 % (0.051±0.016)* |
| NWO | TERRA | **41 %** (0.087±0.090) | **33 %** (0.071±0.142) | - | **26 %** (0.055±0.020) |
| | *AQUA* | *40 % (0.079±0.083)* | *32 % (0.066±0.127)* | *-* | *28 % (0.055±0.020)* |
| SWO | TERRA | **32 %** (0.071±0.070) | **42 %** (0.097±0.257) | - | **26 %** (0.058±0.018) |
| | *AQUA* | *32 % (0.093±0.288)* | *41 % (0.072±0.080)* | *-* | *27 % (0.059±0.018)* |
| NEO | TERRA | **48 %** (0.098±0.094) | **28 %** (0.061±0.144) | - | **24 %** (0.050±0.016) |
| | *AQUA* | *46 % (0.086±0.082)* | *28 % (0.057±0.115)* | *-* | *26 % (0.050±0.016)* |
| SEO | TERRA | **36 %** (0.079±0.070) | **39 %** (0.087±0.224) | - | **25 %** (0.055±0.016) |
| | *AQUA* | *36 % (0.075±0.071)* | *38 % (0.080±0.217)* | *-* | *26 % (0.055±0.016)* |




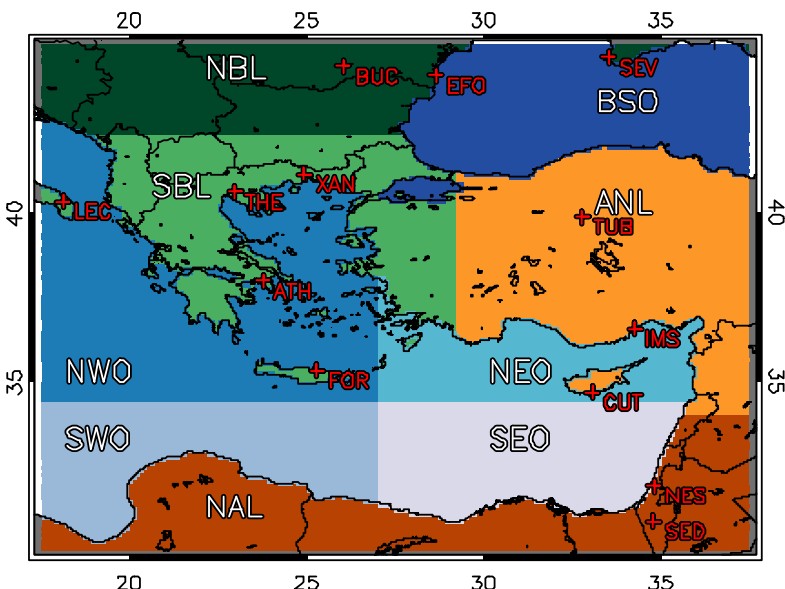

**Figure 1.** Eastern Mediterranean map with the 9 sub-regions selected for the generalization of
our results and the location of the AERONET stations used for the validation of MODIS
satellite data. The 9 sub-regions are: NBL (Northern Balkans Land), SBL (Southern Balkans
Land), ANL (Anatolia Land), NAL (Northern Africa Land), BSO (Black Sea Oceanic), NWO
(North-Western Oceanic), SWO (South-Western Oceanic), NEO (North-Eastern Oceanic) and
SEO (South-Eastern Oceanic). The full names and the geolocation of the 13 AERONET
stations appearing in the map are available in Table 1.

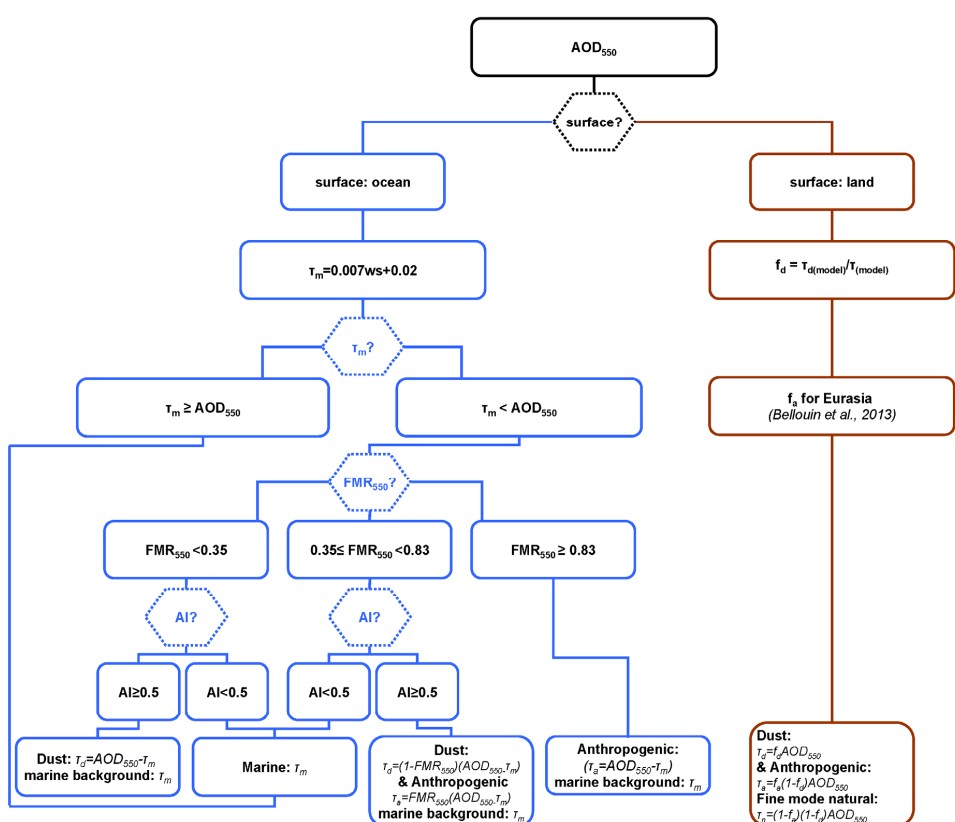

**Figure 2.** Flowchart with the methodology followed for the calculation of the anthropogenic
aerosol, dust and marine aerosol optical depths ($\tau_a$, $\tau_d$ and $\tau_m$) over the sea (blue color) and the
anthropogenic aerosol, dust and fine mode natural aerosol optical depths ($\tau_a$, $\tau_d$ and $\tau_n$) over
land (brown color).

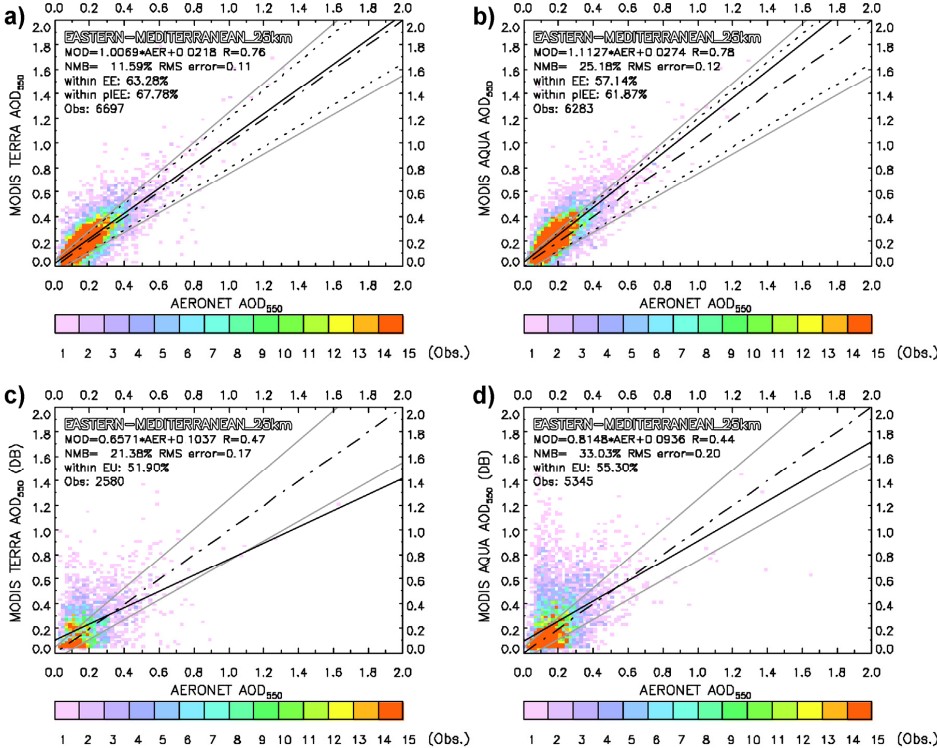

**Figure 3.** Comparison of spatially (using a 25 x 25 km$^2$ window around each station) and
temporally (±30 min from the MODIS overpass time) collocated MODIS Collection 051
level-2 and AERONET sunphotometric (quadratically interpolated) AOD$_{550}$ observations for
the Eastern Mediterranean stations: (a) for MODIS Terra DT data, (b) for MODIS Aqua DT
data, (c) for MODIS Terra DB data and (d) for MODIS Aqua DB data. The color scale
corresponds to the number of MODIS-AERONET collocation points that fall within 0.02 x
0.02 grid boxes. The solid line is the regression line of the MODIS-AERONET observations,
the dashed-dotted line is the 1:1 line, the dotted lines represent the expected error (EE)
envelope and the grey lines the pre-launch expected error (plEE) envelope (Expected
Uncertainty - EU envelope for DB data). The slope and the intercept of the regression line, the
correlation coefficient R, the normalized mean bias (NMB), the root mean squared (RMS)
error, the percentage of the collocation points that fall within the EE and plEE and the number
of all the collocation points.





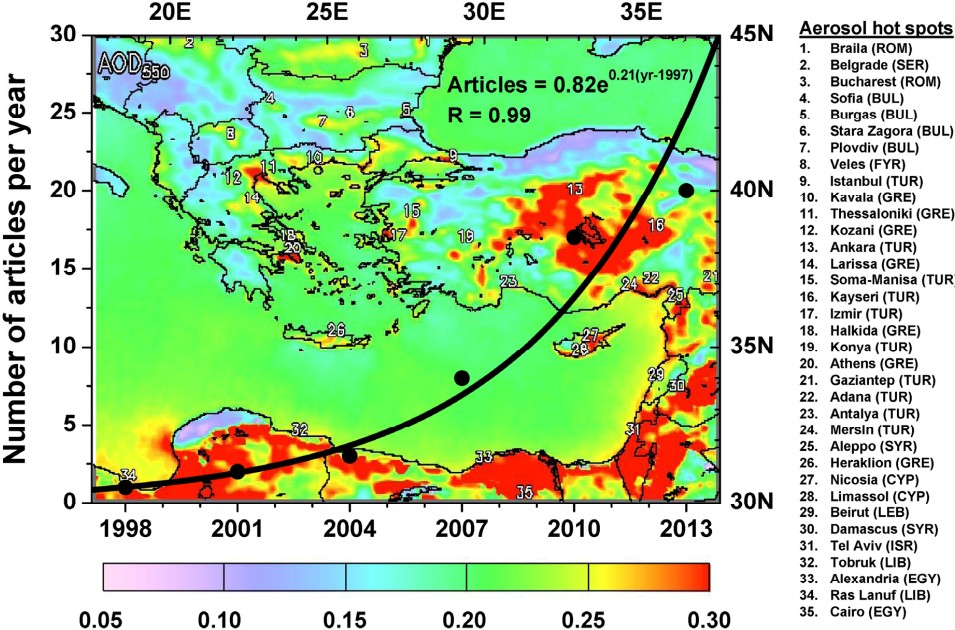

**Figure 4.** $AOD_{550}$ patterns over Eastern Mediterranean as seen by MODIS Terra during the period 3/2000-12/2012 (3/2000-12/2007 for regions of North Africa covered by DB data only). The colorscale corresponds to the $AOD_{550}$ levels while the top x-axis and the right y-axis correspond to the longitude (°E) and latitude (°N), respectively. The position of 35 aerosol hot spots is marked on the map (numbers from 1 to 35) while the names of the places and the countries where the hot spots are located appear on the right of the map. In the same figure the exponential growth of the number of satellite-based articles focusing on aerosols over the greater Eastern Mediterranean from 1997 to 2014 is shown (black line). The black dots represent the number of articles published within three year intervals. The bottom x-axis and the left y-axis correspond to the years and the number of published articles, respectively. The exponential growth corresponds to a near doubling of the publication rate every 3 years.



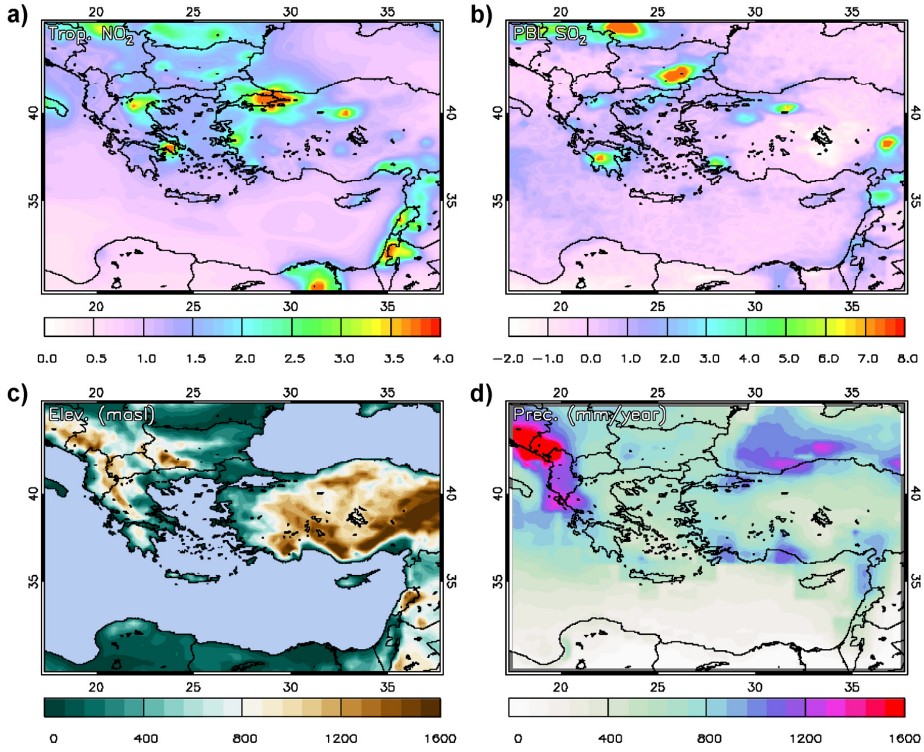

**Figure 5.** (a) Tropospheric $NO_2$ levels and (b) Planetary boundary layer $SO_2$ levels (in $10^{15}$
molecules/cm$^2$) over the Eastern Mediterranean as seen from OMI/AURA (2005-2012), (c)
Topography (GTOPO elevation data in meters above sea level) and (d) Annual precipitation
levels (in mm/year) from 3B43 TRMM and Other Sources Monthly Rainfall Product (2000-

7    2012).



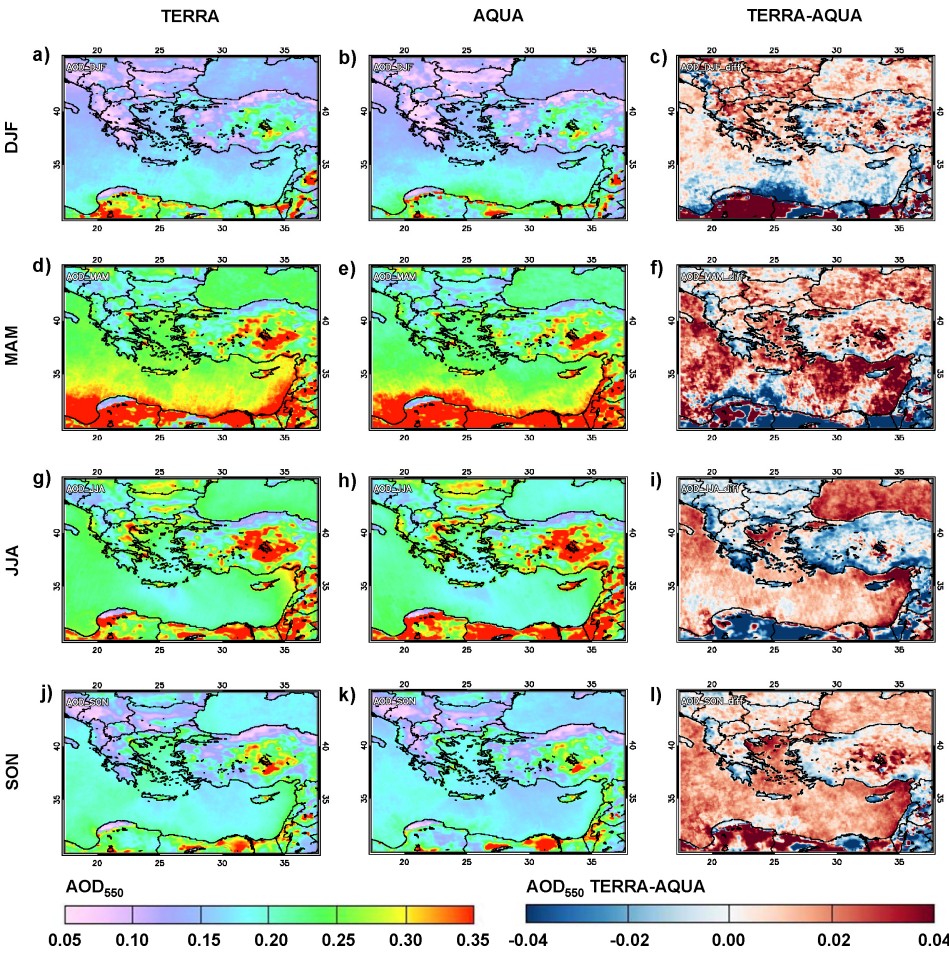

**Figure 6.** Seasonal AOD$_{550}$ patterns over the Eastern Mediterranean as seen by MODIS Terra

(left column) during the period 3/2000-12/2012 (3/2000-12/2007 for regions of North Africa

covered by DB data only) and MODIS Aqua (middle column) during the period 7/2002-

12/2012. The differences between MODIS Terra and Aqua AOD$_{550}$ on a seasonal basis appear

on the right column.





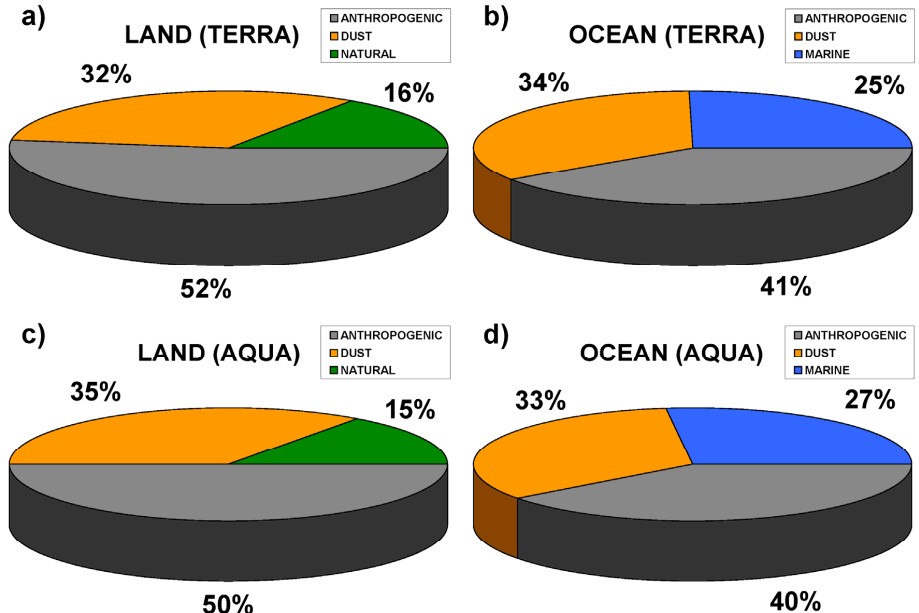

**Figure 7.** Relative contribution of anthropogenic aerosols, dust and fine mode natural
aerosols to the total $AOD_{550}$ over the land covered part of Eastern Mediterranean based on
MODIS Terra (a) and MODIS Aqua (c) observations and relative contribution of
anthropogenic aerosols, dust and marine aerosols to the total $AOD_{550}$ over the oceanic part of
Eastern Mediterranean based on MODIS Terra (b) and MODIS Aqua (d) observations.





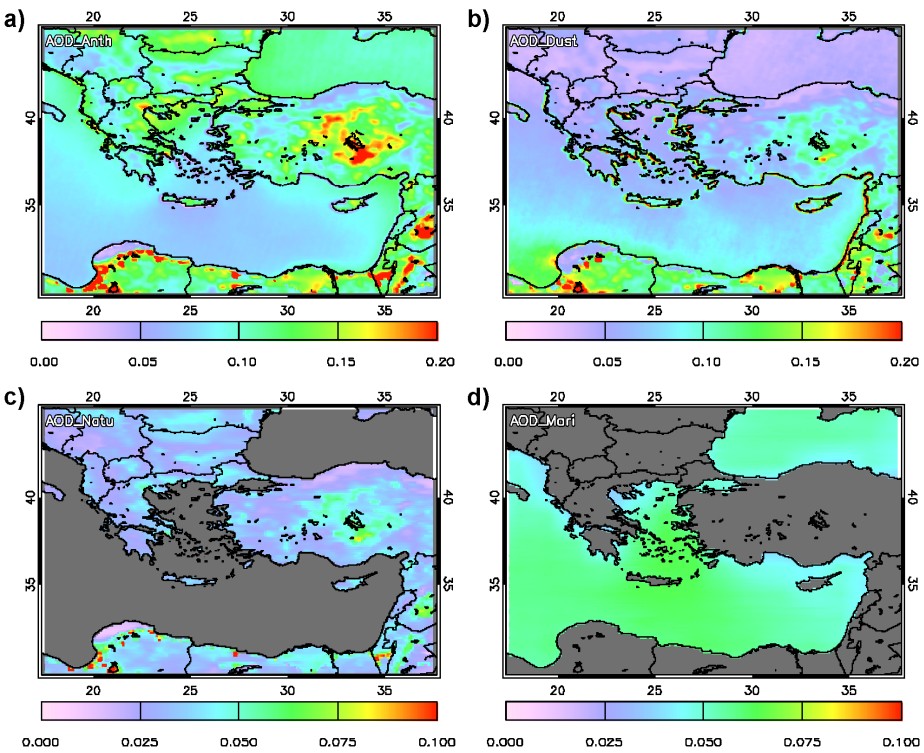

**Figure 8.** (a) Anthropogenic aerosol ($\tau_a$), (b) dust ($\tau_d$), (c) fine mode natural aerosol ($\tau_n$) and (d) marine aerosol ($\tau_m$) patterns over the Eastern Mediterranean based on MODIS Terra observations during the period 3/2000-12/2012 (3/2000-12/2007 for regions of North Africa covered by DB data only).



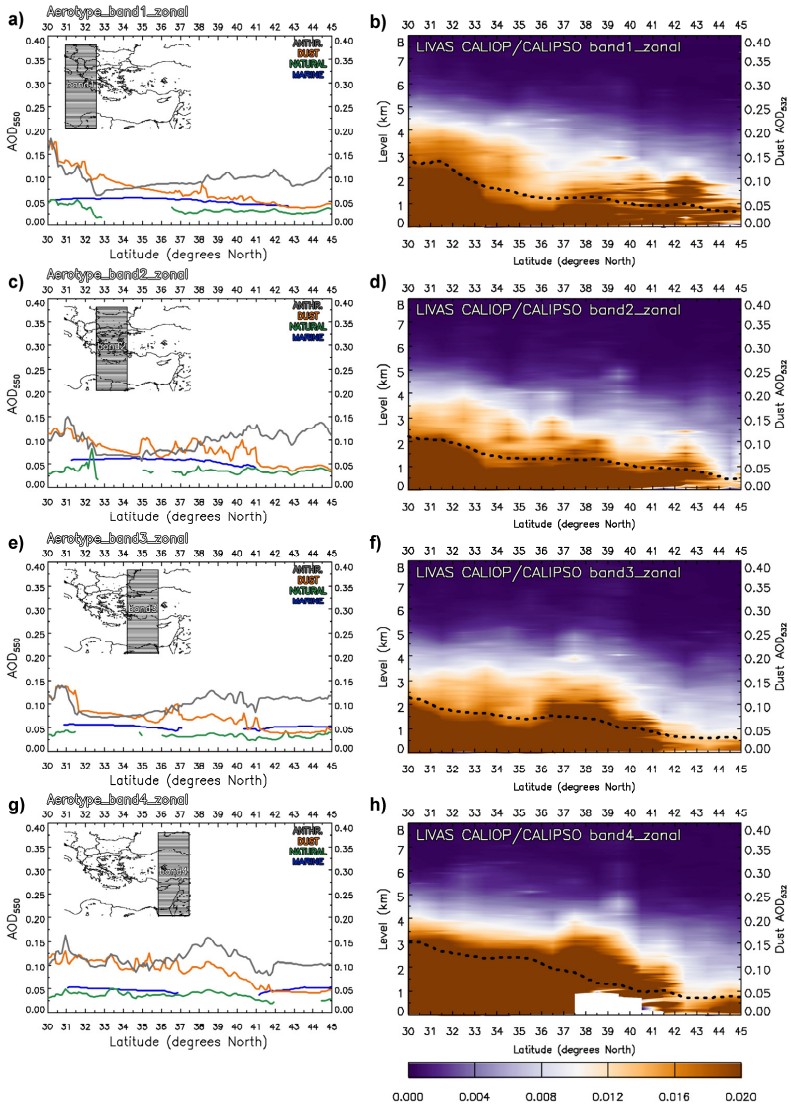

**Figure 9.** Left column: Latitudinal variability of anthropogenic aerosols ($\tau_a$), dust ($\tau_d$), fine mode natural aerosols ($\tau_n$) and marine aerosols ($\tau_m$) for four 5-degree longitudinal bands (see embedded maps) covering Eastern Mediterranean based on MODIS Terra observations. Right column: Latitudinal variability of dust extinction coefficients at 532 nm in km$^{-1}$ (colorscale corresponds to the extinction coefficients and left y-axis to the atmospheric levels) and dust aerosol optical depth at 532 nm (dotted line corresponding to the right-y-axis) for the same four bands from LIVAS CALIOP/CALIPSO observations.



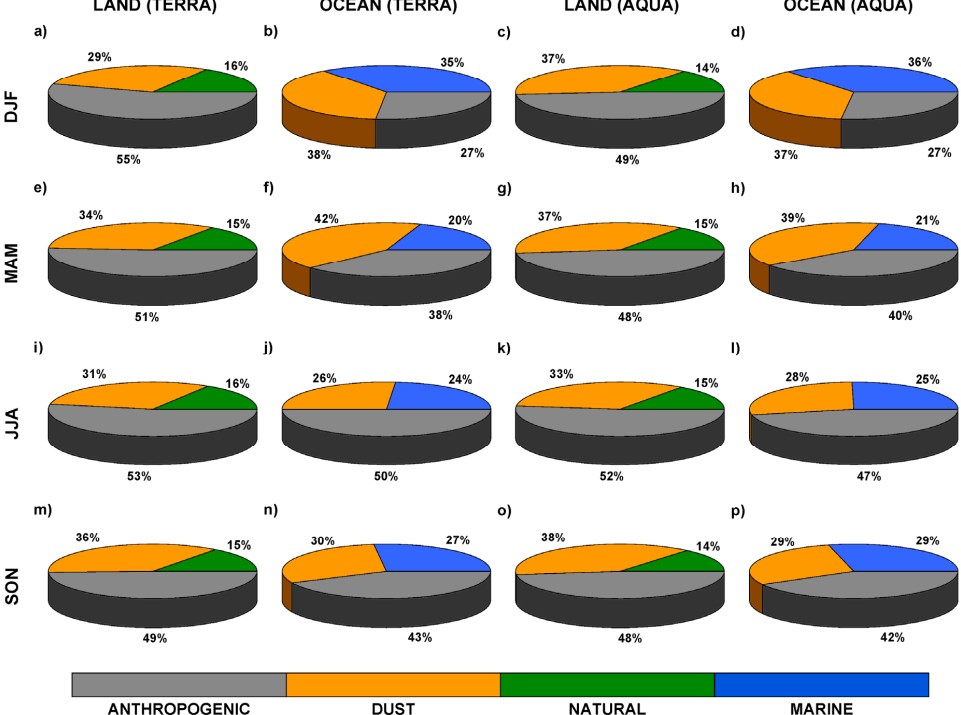

**Figure 10.** Seasonal relative contribution of anthropogenic aerosols, dust and fine mode
natural aerosols to the total $AOD_{550}$ over the land covered part of Eastern Mediterranean
based on MODIS Terra (a, e, i, m) and MODIS Aqua (c, g, k, o) observations and seasonal
relative contribution of anthropogenic aerosols, dust and marine aerosols to the total $AOD_{550}$
over the oceanic part of Eastern Mediterranean based on MODIS Terra (b, f, j, n) and MODIS
Aqua (d, h, i, p) observations.





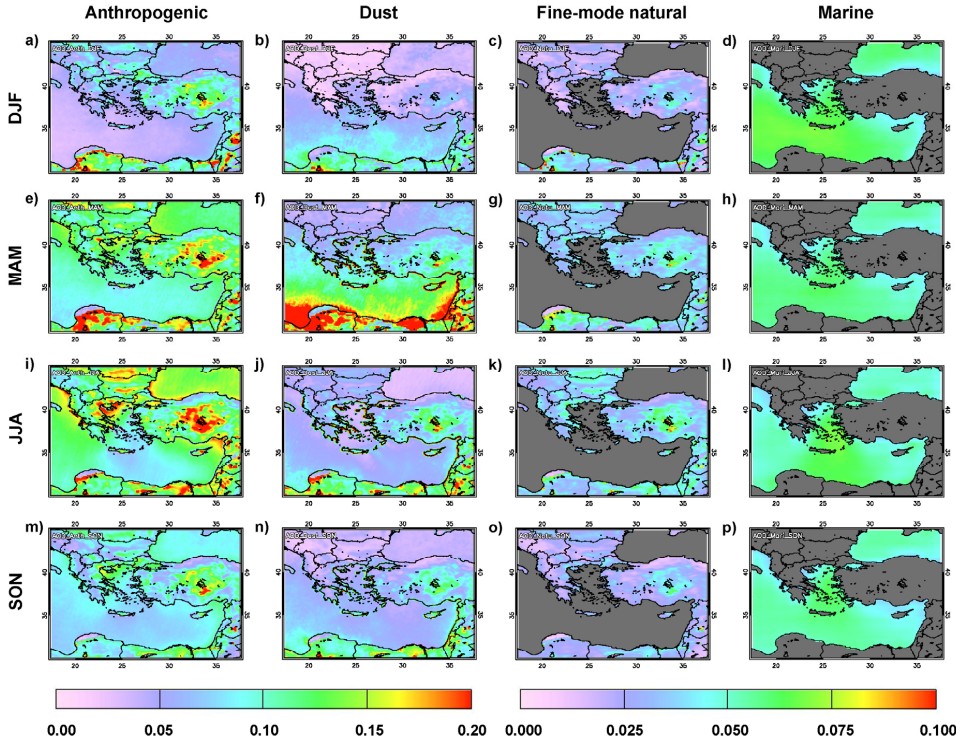

**Figure 11.** Seasonal (a, e, i, m) anthropogenic aerosol ($\tau_a$), (b, f, j, n) dust ($\tau_d$), (c, g, k, o) fine mode natural aerosol ($\tau_n$) and (d, h, i, p) marine aerosol ($\tau_m$) patterns over the Eastern Mediterranean based on MODIS Terra observations during the period 3/2000-12/2012 (3/2000-12/2007 for regions of North Africa covered by DB data only).



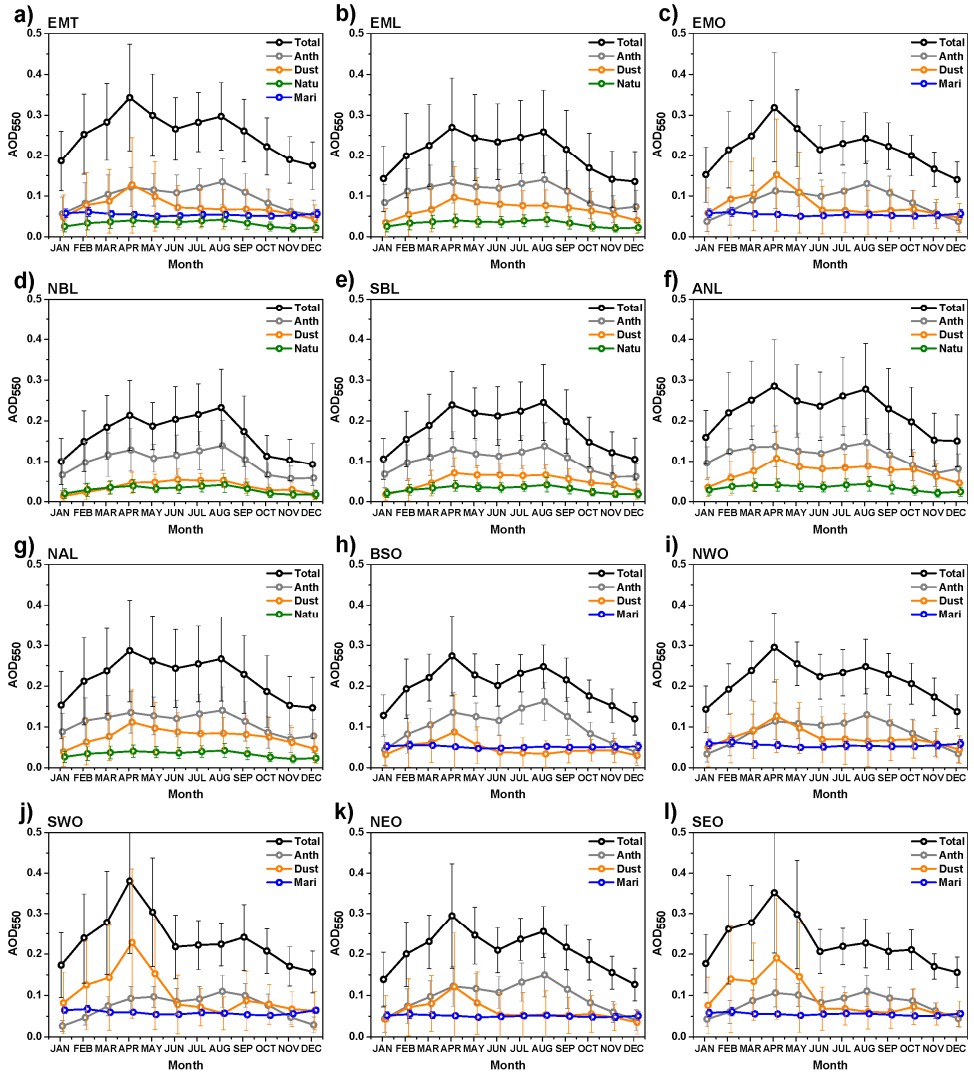

**Figure 12.** Seasonal variability of anthropogenic aerosols ($\tau_a$), dust ($\tau_d$), fine mode natural
aerosols ($\tau_n$) and marine aerosols ($\tau_m$) over Eastern Mediterranean (EMT), over the land
covered part (EML), over the oceanic part (EMO) and over the 9 sub-regions of the Eastern
Mediterranean appearing in Fig. 1 based on MODIS Terra observations. The error bars
represent the $\pm 1\sigma$ values calculated from monthly gridded data.