# Peer review of "Spatiotemporal variability and contribution of different aerosol"

_Atmospheric Chemistry and Physics, 2016_

## Referee Comment (RC1) · Anonymous Referee #1 · 9 Aug 2016

This study characterizes the spatiotemporal variability and relative contribution of different types of aerosols to the Aerosol Optical Depth (AOD) over the Eastern Mediterranean as derived from MODIS satellite data. I agree this paper published in ACP just with some minor revisions. 1. Line 1, Page 2: AERONET provide the full name. 2. Line 3, Page 3: "Lelieveld et al., 2002, Giorgi, 2006" Should be "Lelieveld et al., 2002; Giorgi, 2006)." 3. Line 7, Page 3: Please add some references for the sentence "The Mediterranean is also recognized as a crossroads between three continents where aerosols of various types accumulate". 4. Line 1, Page 4: The references format should be unified, for example: "Gkikas et al., 2013; 2014;" should be "Gkikas et al., 2013, 2014".,
etc. Please revise all similar reference formats through the full text. 5. Line 7, Page 7: "AE440-870"should be "AE440-870". 6. Line 11, Page 8: "resolution of 1o x 1.25o ": Please add the note which is latitude? Which is longitude? 7. Line 32, Page 8: What is the resolution of wind speed data at 10 m above surface from ERA-Interim reanalysis used in your paper? 8. Why have the years after 2012 been excluded? 9. AERONET AOD550 is how calculated? 10. Line 12-13, Page 21: Please add some references for the sentence "Over the ocean, a profound maximum is observed in spring which is due to the well documented transport of significant amounts of dust from the Sahara Desert extending across the North African coast and the ocean." 11. Line 3, Page 25: "from 49 % in SON to 50 %" should be "from 49 % in SON to 55 %". 12. Figure 6d: AOD550 at NAL region in spring is bigger than it in summer. Why? 13. The conclusion is a very long summary, whether can be shorten appropriately? 14. Whether to consider joining the back trajectory model to track the source of dust?

---

## Referee Comment (RC2) · Anonymous Referee #2 · 31 Aug 2016

The authors describe the spatial and temporal variation of aerosols over the eastern Mediterranean, using a variety of satellite and ground-based (AERONET) observations together with model results. Contributions of several aerosol types to the total AOD are analyzed. This is a very good paper describing how different sources of information can be used together and I recommend publication in ACP with only very few minor clarifications and editorial suggestions. Minor comments

Overall, suggest to: insert 'the' before 'Mediterranean' throughout the whole MS replace 'ocean' with 'sea' 6, 33 'such as' > 'such that' 7, 13 Li et al (2013) is missing from

the references 7, 27-28 needs editing 8, section 2.4: give the definition of the AAI (I assume you mean the absorbing aerosol index) and explain that is a qualitative measure for aerosol absorption, give a reference 8, 25 'is applied on spectral measurements from both TOMS and OMI sensor' > 'is applied to spectral measurements from both the TOMS and OMI sensors' 14, 5 is there a reason for these large overestimations of sea salt contribution to the AOD? 17, 21 is it useful to give both the EE and pIEE? They are not very different and in principle do not provide additional info for this kind of study, or does it? 18, 10 'about' > 'for the' 18, 16 why don't you use the same area as for DT? 18, 18 25x25 19,22 taking this into account 20, 26 and elsewhere throughout the MS (e.g. 30, 22): formulation. Due to the low precipitation (removal) the AOD remains high, but actually the AOD is high because of emissions and atmospheric processes forming aerosol particles 21, 4 same as 20,26: the sources are what they are (important when large) but they do not become more prominent due to limited washout. Rephrase 27, 20 add 'with' after winter month 28, 33 remove 'region of' Figure 2: the text in the boxes is very small: could you make this larger? Figure 3: the headers above the text are difficult to read and not needed since they are all the same; removing them leaves more space to increase the font size of the statistical results given there. In the caption an expected uncertainty is mentioned (line 12) which does not occur in the text. Is this different from the EE, and if so, please explain how evaluated.

---

## Author Response (AR1)

**Response to Reviewers**

The authors would like to thank the Anonymous Reviewers for their comments. Below, please find attached our response to each one of the reviewers' comments:

**Reviewer #1**

1. Line 1, Page 2: AERONET provide the full name.

We have added the AERONET full name in the revised version of the manuscript.

2. Line 3, Page 3: "Lelieveld et al., 2002, Giorgi, 2006" Should be "Lelieveld et al., 2002; Giorgi, 2006)."

This has been corrected in the revised version of the manuscript.

3. Line 7, Page 3: Please add some references for the sentence "The Mediterranean is also recognized as a crossroads between three continents where aerosols of various types accumulate".

The classic paper from Lelieveld et al. (2002) that inserted the term "crossroads" for the Mediterranean Basin has been added.

4. Line 1, Page 4: The references format should be unified, for example: "Gkikas et al., 2013; 2014;" should be "Gkikas et al., 2013, 2014"., etc. Please revise all similar reference formats through the full text.

We have corrected this wherever it appeared in the manuscript.

5. Line 7, Page 7: "AE440-870"should be "$AE_{440-870}$".

This has been corrected in the revised version of the manuscript.

6. Line 11, Page 8: "resolution of $1^{o}$ x $1.25^{o}$ ": Please add the note which is latitude? Which is longitude?

We write it like this "1o (latitude) x 1.25o (longitude) " in the revised version of the manuscript.

7. Line 32, Page 8: What is the resolution of wind speed data at 10 m above surface from ERA-Interim reanalysis used in your paper?

We thank the reviewer for giving use the opportunity to clarify this. We added the following in the revised version of the manuscript "...the data can be acquired at various resolutions (in this work 1ox1o) through ECMWF's website..."

8. Why have the years after 2012 been excluded?

We refer in the revised manuscript that the years after 2012 have been excluded as the MACC reanalysis data are only available for the period 2003-2012.

9. AERONET AOD$_{550}$ is how calculated?

We had already written about this in the paper "... Here, we use quadratic fits on a log-log scale to interpolate the AERONET data (AODs at 440, 500, 675 and 870 nm) to the MODIS band-effective wavelength of 550 nm (Eck et al., 1999; Levy et al., 2010)..."

However, we also added the following phrase in order to make it more clear "...So, we can directly compare the MODIS AOD$_{550}$ retrievals against AERONET observations..."

10. Line 12-13, Page 21: Please add some references for the sentence "Over the ocean, a profound maximum is observed in spring which is due to the well documented transport of significant amounts of dust from the Sahara Desert extending across the North African coast and the ocean."

As this spring transport is found in numerous papers about the area we added the following in the sentence "...(see Barnaba and Gobbi., 2004 and the list of references given in the introduction)..."

11. Line 3, Page 25: "from 49 % in SON to 50 %" should be "from 49 % in SON to 55 %".

This has been corrected in the revised version of the manuscript.

12. Figure 6d: AOD$_{550}$ at NAL region in spring is bigger than it in summer. Why?

This is mostly due to the significant amount of dust produced and transported in this region during spring. This can also be seen in Fig. 11 where we see the significantly higher amounts of dust over the region in spring compared to summer.

13. The conclusion is a very long summary, whether can be shorten appropriately?

We agree with the reviewer that the conclusions section is pretty long. So we decided to rename this section to Summary and conclusions. As this is a long paper and it is very common among scientists to focus on the abstract and the conclusions we prefer to keep the Summary and conclusions section as informative as possible (numbers, conclusions, methodology, future use of the dataset, etc.).

14. Whether to consider joining the back trajectory model to track the source of dust?

We thank the reviewer for his/her comment. We have already plans to extend the methodology as discussed in the Conclusions using more recent datasets that will allow the extension of the timeseries and we also are working on possible improvements of the methodology. Indeed the use of trajectory models could be useful especially over the sea as over land the method relies on data from models and the trajectories are not expected to return significantly different results. One of our future goals is to separate cases that dust is transported from Sahara from case that dust comes from Middle East and trajectory analysis will play a critical role.

**Reviewer #2**

1. insert 'the' before 'Mediterranean' throughout the whole MS.

We have addressed this in the revised version of the manuscript.

2. replace 'ocean' with 'sea'.

We corrected this.

3. 6, 33 'such as' > 'such that'.

We corrected this within the manuscript. We also changed this on Figs. 7 and 10.

4. 7, 13 Li et al (2013) is missing from the references

The reference has been added.

5. 7, 27-28 needs editing.

This sentence has been rephrased to "...This correction results to an improvement of the AOD532 product. Comparison against spatially and temporally co-located AERONET observations (Amiridis et al., 2013) returned an absolute bias of ~ -0.03..."

6. 8, section 2.4: give the definition of the AAI (I assume you mean the absorbing aerosol index) and explain that is a qualitative measure for aerosol absorption, give a reference.

We thank the reviewer for giving us the opportunity to clarify this. We mention in the revised version of the manuscript that this is the UV Aerosol Index (AI) also known as Absorbing Aerosol index and it is a qualitative indicator of the presence of UV absorbing aerosols in the atmosphere such as biomass burning and dust. We added the following reference: (Torres et al., 1998) and we also mention that positive AI values generally represent absorbing aerosols while small or negative values represent non-absorbing aerosols.

7. 8, 25 'is applied on spectral measurements from both TOMS and OMI sensor' > 'is applied to spectral measurements from both the TOMS and OMI sensors'.

We corrected this.

8. 14, 5 is there a reason for these large overestimations of sea salt contribution to the AOD?

Probably it has to do with the use of Smirnov (2003) formula for the calculation of marine AOD. It works well for open seas where we have stronger winds and sea salt dominates (when other types of aerosols dominate usually their AODs are pretty high, e.g. Saharan dust over the Atlantic Ocean). So, when the formula returns values higher than the MODIS AOD all the AOD is attributed to sea salt which is OK for marine aerosol dominated regions. However, for the Mediterranean Basin (closed sea and not an open ocean) where other types of aerosols dominate the formula leads to a very frequent attribution of the total AOD to sea salt. Consequently this leads to a significant overestimation of marine aerosols. As there is already enough discussion on the tests we did in order to fine-tune the algorithm we prefer not to get into details about the reason of the overestimations.

We also have to mention that we mistakenly wrote in the original version of the paper that we used the 0.5 critical value for AI. We used this value in our tests but in the end we preferred to use 1 which is in accordance to other studies in the area. We have corrected Fig. 2 accordingly.

9. 17, 21 is it useful to give both the EE and pIEE? They are not very different and in principle do not provide additional info for this kind of study, or does it?

We believe yes. We wanted to show that even though in some cases the MODIS data have a lower percentage than 66% within the EE (the EE is a result of global validation studies) can still have a percentage within the plEE which was defined encompassing the sum of absolute and relative AOD errors prior to the launch of Terra (Kaufman et al., 1997).

10. 18, 10 'about' > 'for the'.

We corrected this.

11. 18, 16 why don't you use the same area as for DT?

Basically, we use the same spatial window for the validation ($25 \times 25$ km$^2$) for both DT and DB. In order to be in line with previous studies using Col. 5 MODIS AOD$_{550}$ from DT and DB we also did a DT validation using the 50 km window as in Levy et al. (2010) and a DB validation using high quality retrievals and a 25 km window as in e.g. Shi et al. (2011). In the case of DB we also wanted to show that the use of all the data instead of high quality data only did not change the correlation with the ground-based data significantly but diminished the statistical sample by a factor of 5.

12. 18, 18 25x25.

We corrected this.

13. 19,22 taking this into account

We corrected this.

14. 20, 26 and elsewhere throughout the MS (e.g. 30, 22): formulation. Due to the low precipitation (removal) the AOD remains high, but actually the AOD is high because of emissions and atmospheric processes forming aerosol particles

We thank the reviewer for giving us the opportunity to correct this in the revised manuscript. We added the following in 20,26: "...It has to be highlighted that the AOD550 over these regions is high primarily due to the emissions and the atmospheric processes forming aerosol particles. The low removal rates from precipitation just preserve the $AOD_{550}$ levels high..." and in 30,22: "...Precipitation is the major washout mechanism of atmospheric pollutants., Low removal rates from precipitation contribute in preserving high the $AOD_{550}$ levels which are a result of emissions and other atmospheric processes..."

15. 21, 4 same as 20,26: the sources are what they are (important when large) but they do not become more prominent due to limited washout.

We rephrased the sentence to: "...The limited washout by precipitation (see also Papadimas et al., 2008) and also the enhanced photochemical production of secondary organic aerosols (Kanakidou et al., 2011 and references therein) contribute to the high AODs appearing over local sources..."

16. Rephrase 27, 20 add 'with' after winter month.

We corrected this.

17. 28, 33 remove 'region of'.

We corrected this.

18. Figure 2: the text in the boxes is very small: could you make this larger?

We did them larger where possible (all the boxes except for the last line). We will keep the figure as big as possible in the final paper.

19. Figure 3: the headers above the text are difficult to read and not needed since they are all the same; removing them leaves more space to increase the font size of the statistical results given there. In the caption an expected uncertainty is mentioned (line 12) which does not occur in the text. Is this different from the EE, and if so, please explain how evaluated.

We agree with the reviewer, we have revised the Figure accordingly. The expected Uncertainty refers to the MODIS DB data and is given in Section 4.1 "... The MODIS Terra DB data overestimate $AOD_{550}$ by 21.38 % (NMB) with 51.90 % of the data falling within the expected uncertainty (EU) envelope assuming a DB expected uncertainty of $\pm 0.05 \pm 20\% AOD_{AERONET}$ (Hsu et al., 2006)..."

**Main document changes and comments**

| Page 2: Deleted | User | 9/28/2016 4:35:00 PM |
|---|---|---|

AERONET

| Page 2: Inserted | User | 9/28/2016 4:35:00 PM |
|---|---|---|

from the AErosol RObotic NETwork (AERONET)

| Page 2: Deleted | User | 9/27/2016 11:31:00 PM |
|---|---|---|

oceanic

| Page 2: Inserted | User | 9/27/2016 11:31:00 PM |
|---|---|---|

sea

| Page 3: Deleted | User | 9/28/2016 4:32:00 PM |
|---|---|---|

,

| Page 3: Inserted | User | 9/28/2016 4:32:00 PM |
|---|---|---|

;

| Page 3: Inserted | User | 9/28/2016 4:57:00 PM |
|---|---|---|

(Lelieveld et al., 2002)

| Page 3: Deleted | User | 9/30/2016 12:39:00 PM |
|---|---|---|

the

| Page 3: Deleted | User | 9/30/2016 12:41:00 PM |
|---|---|---|

applied

| Page 3: Inserted | User | 9/30/2016 12:41:00 PM |
|---|---|---|

used

| Page 3: Deleted | User | 9/30/2016 12:43:00 PM |
|---|---|---|

al.

| Page 4: Deleted | User | 9/28/2016 4:39:00 PM |
|---|---|---|

;

| Page 4: Inserted | User | 9/28/2016 4:39:00 PM |
|---|---|---|

,

| Page 4: Deleted | User | 9/28/2016 4:39:00 PM |
|---|---|---|

;

| Page 4: Inserted | User | 9/28/2016 4:39:00 PM |
|---|---|---|

,

| Page 4: Deleted | User | 9/28/2016 4:39:00 PM |

;
,

| Page 4: Inserted | User | 9/28/2016 4:39:00 PM |

,

| Page 4: Deleted | User | 9/28/2016 4:39:00 PM |

;
,

| Page 4: Inserted | User | 9/28/2016 4:39:00 PM |

,

| Page 4: Deleted | User | 9/28/2016 4:39:00 PM |

;
,

| Page 4: Inserted | User | 9/28/2016 4:39:00 PM |

,

| Page 4: Deleted | User | 9/28/2016 4:39:00 PM |

;
,

| Page 4: Inserted | User | 9/28/2016 4:39:00 PM |

,

| Page 4: Deleted | User | 9/28/2016 4:39:00 PM |

;
,

| Page 4: Inserted | User | 9/28/2016 4:39:00 PM |

,

| Page 4: Deleted | User | 9/28/2016 4:41:00 PM |

;
,

| Page 4: Inserted | User | 9/28/2016 4:41:00 PM |

,

| Page 4: Inserted | User | 9/27/2016 11:18:00 PM |

the

| Page 5: Deleted | User | 9/27/2016 11:32:00 PM |

ocean

| Page 5: Inserted | User | 9/27/2016 11:32:00 PM |

sea

| Page 5: Inserted | User | 9/27/2016 11:18:00 PM |
|---|---|---|

the

| Page 6: Deleted | User | 9/27/2016 11:32:00 PM |
|---|---|---|

ocean

| Page 6: Inserted | User | 9/27/2016 11:32:00 PM |
|---|---|---|

sea

| Page 6: Deleted | User | 9/27/2016 11:32:00 PM |
|---|---|---|

ocean

| Page 6: Inserted | User | 9/27/2016 11:32:00 PM |
|---|---|---|

sea

| Page 6: Deleted | User | 9/27/2016 11:33:00 PM |
|---|---|---|

ocean

| Page 6: Inserted | User | 9/27/2016 11:33:00 PM |
|---|---|---|

sea

| Page 6: Inserted | User | 9/27/2016 11:19:00 PM |
|---|---|---|

the

| Page 6: Inserted | User | 9/27/2016 11:19:00 PM |
|---|---|---|

the

| Page 6: Deleted | User | 9/27/2016 11:30:00 PM |
|---|---|---|

as

| Page 6: Inserted | User | 9/27/2016 11:30:00 PM |
|---|---|---|

that

| Page 7: Inserted | User | 9/30/2016 12:02:00 AM |
|---|---|---|

So, we can directly compare the MODIS $AOD_{550}$ retrievals against AERONET observations.

| Page 7: Formatted | User | 9/30/2016 12:02:00 AM |
|---|---|---|

Subscript

| Page 7: Formatted | User | 9/28/2016 4:42:00 PM |
|---|---|---|

Subscript

| Page 7: Deleted | User | 9/27/2016 11:50:00 PM |
|---|---|---|

using

| Page 7: Inserted | User | 9/27/2016 11:50:00 PM |
|---|---|---|

applying

| Page 7: Deleted | User | 9/27/2016 11:51:00 PM |
|---|---|---|

is applied

| Page 7: Deleted | User | 9/27/2016 11:51:00 PM |
|---|---|---|

improved

| Page 7: Inserted | User | 9/27/2016 11:51:00 PM |
|---|---|---|

improvement of the $AOD_{532}$ product. Comparison

| Page 7: Formatted | User | 9/27/2016 11:51:00 PM |
|---|---|---|

Subscript

| Page 7: Deleted | User | 9/28/2016 12:04:00 AM |
|---|---|---|

$AOD_{532}$ absolute bias of ~ -0.03 compared

| Page 7: Inserted | User | 9/28/2016 12:02:00 AM |
|---|---|---|

against

| Page 7: Deleted | User | 9/28/2016 12:02:00 AM |
|---|---|---|

to

| Page 7: Inserted | User | 9/28/2016 12:02:00 AM |
|---|---|---|

 returned an absolute bias of ~ -0.03

| Page 8: Inserted | User | 9/28/2016 12:07:00 AM |
|---|---|---|

UV

| Page 8: Inserted | User | 9/28/2016 4:44:00 PM |
|---|---|---|

 (latitude)

| Page 8: Inserted | User | 9/28/2016 4:44:00 PM |
|---|---|---|

 (longitude)

| Page 8: Inserted | User | 9/28/2016 12:15:00 AM |
|---|---|---|

 (also known as Absorbing Aerosol index)

| Page 8: Formatted | User | 9/28/2016 12:15:00 AM |
|---|---|---|

English (U.K.)

| Page 8: Deleted | User | 9/28/2016 12:28:00 AM |
|---|---|---|

 (e.g.

| Page 8: Inserted | User | 9/28/2016 12:28:00 AM |
|---|---|---|

such as

| Page 8: Deleted | User | 9/28/2016 12:28:00 AM |
|---|---|---|

,

| Page 8: Inserted | User | 9/28/2016 12:28:00 AM |
|---|---|---|

and

| Page 8: Deleted | User | 9/28/2016 12:28:00 AM |
|---|---|---|

).

| Page 8: Inserted | User | 9/28/2016 12:28:00 AM |
|---|---|---|

 (Torres et al., 1998). Positive AI values generally represent absorbing aerosols while small or negative values represent non-absorbing aerosols.

| Page 8: Deleted | User | 9/28/2016 12:16:00 AM |
|---|---|---|

on

| Page 8: Inserted | User | 9/28/2016 12:16:00 AM |
|---|---|---|

to

| Page 9: Inserted | User | 9/28/2016 4:49:00 PM |
|---|---|---|

(in this work $1^{o}x1^{o}$)

| Page 9: Formatted | User | 9/28/2016 4:49:00 PM |
|---|---|---|

Superscript

| Page 9: Formatted | User | 9/28/2016 4:49:00 PM |
|---|---|---|

Superscript

| Page 9: Inserted | User | 9/27/2016 11:19:00 PM |
|---|---|---|

the

| Page 9: Inserted | User | 9/28/2016 5:16:00 PM |
|---|---|---|

 for the period 2003-2012

| Page 11: Deleted | User | 9/27/2016 11:33:00 PM |
|---|---|---|

ocean

| Page 11: Inserted | User | 9/27/2016 11:33:00 PM |
|---|---|---|

sea

| Page 11: Deleted | User | 9/30/2016 12:59:00 PM |
|---|---|---|

at

| Page 11: Inserted | User | 9/30/2016 12:59:00 PM |
|---|---|---|

to

| Page 12: Inserted | User | 9/27/2016 11:19:00 PM |
|---|---|---|

the

| Page 13: Deleted | User | 9/27/2016 11:34:00 PM |
|---|---|---|

**Ocean**

| | | |
|---|---|---|
| **Page 13: Inserted** | **User** | **9/27/2016 11:34:00 PM** |

**Sea**

| | | |
|---|---|---|
| **Page 13: Deleted** | **User** | **9/27/2016 11:34:00 PM** |

ocean

| | | |
|---|---|---|
| **Page 13: Inserted** | **User** | **9/27/2016 11:34:00 PM** |

sea

| | | |
|---|---|---|
| **Page 13: Deleted** | **User** | **9/27/2016 11:34:00 PM** |

 regions

| | | |
|---|---|---|
| **Page 13: Inserted** | **User** | **9/27/2016 11:34:00 PM** |

the

| | | |
|---|---|---|
| **Page 13: Deleted** | **User** | **9/27/2016 11:34:00 PM** |

ocean

| | | |
|---|---|---|
| **Page 13: Inserted** | **User** | **9/27/2016 11:34:00 PM** |

sea

| | | |
|---|---|---|
| **Page 13: Inserted** | **User** | **9/27/2016 11:38:00 PM** |

the

| | | |
|---|---|---|
| **Page 14: Inserted** | **User** | **9/27/2016 11:20:00 PM** |

the

| | | |
|---|---|---|
| **Page 14: Deleted** | **User** | **9/28/2016 12:57:00 AM** |

critical values

| | | |
|---|---|---|
| **Page 14: Inserted** | **User** | **9/28/2016 12:58:00 AM** |

equations and critical values

| | | |
|---|---|---|
| **Page 14: Deleted** | **User** | **9/28/2016 12:51:00 AM** |

use of the original critical values

| | | |
|---|---|---|
| **Page 14: Inserted** | **User** | **9/28/2016 12:51:00 AM** |

method

| | | |
|---|---|---|
| **Page 14: Inserted** | **User** | **9/27/2016 11:20:00 PM** |

the

| | | |
|---|---|---|
| **Page 14: Deleted** | **User** | **9/27/2016 11:39:00 PM** |

ocean

| Page 14: Inserted | User | 9/27/2016 11:39:00 PM |
|---|---|---|

sea

| Page 14: Inserted | User | 9/27/2016 11:20:00 PM |
|---|---|---|

the

| Page 14: Deleted | User | 9/27/2016 11:39:00 PM |
|---|---|---|

ocean

| Page 14: Inserted | User | 9/27/2016 11:39:00 PM |
|---|---|---|

sea

| Page 14: Inserted | User | 9/27/2016 11:20:00 PM |
|---|---|---|

the

| Page 14: Inserted | User | 9/28/2016 1:08:00 AM |
|---|---|---|

n

| Page 14: Deleted | User | 9/28/2016 12:59:00 AM |
|---|---|---|

0.5

| Page 14: Inserted | User | 9/28/2016 12:59:00 AM |
|---|---|---|

| Page 14: Deleted | User | 9/28/2016 12:59:00 AM |
|---|---|---|

better

| Page 14: Inserted | User | 9/28/2016 12:59:00 AM |
|---|---|---|

well

| Page 14: Inserted | User | 9/27/2016 11:20:00 PM |
|---|---|---|

the

| Page 14: Deleted | User | 9/28/2016 1:07:00 AM |
|---|---|---|

As

| Page 14: Inserted | User | 9/28/2016 1:07:00 AM |
|---|---|---|

The results did not change significantly when using

| Page 14: Deleted | User | 9/28/2016 1:07:00 AM |
|---|---|---|

discussed in Jones and Christopher (2011),

| Page 14: Inserted | User | 9/28/2016 1:07:00 AM |
|---|---|---|

other

| Page 14: Deleted | User | 9/28/2016 1:07:00 AM |
|---|---|---|

the

| Page 14: Inserted | User | 9/28/2016 1:07:00 AM |

s

| Page 14: Deleted | User | 9/28/2016 1:07:00 AM |

of

| Page 14: Inserted | User | 9/28/2016 1:07:00 AM |

(e.g.

[revised manuscript text omitted]

---

## Author Response (AR2)

**Response to the Editor**

The authors would like to thank the Editor for his comments. Below, please find attached our response to each one of the Editor's minor comments:

1. In the first paragraph of page 4 there is a long list of publications dealing with Mediterranean and satellite use. This collection of papers is based simply on satellite use. I would suggest since you are not commenting on what anyone of these authors really did and since this is not an overview paper, to limit the publications on this section based on the importance of their results.

We thank the Editor for this comment. We agree with the Editor that this is not a review paper as it discusses new findings. On the other hand our goal was to submit a kind of comprehensive reference study for the area which could be used not only for scientific but also for educational purposes. The main idea behind giving all these studies is that 1. the studies are separated according to the satellite instruments they use and 2. the statistics given in Fig. 4 are produced using these references. If we removed some of them, then Fig. 4 which shows the importance of the region and the increasing interest about it from the scientific community would not be valid any more.

Therefore, we kindly ask from the Editor to allow us keeping these references in the text.

2. There are several papers comparing aeronet stations with MODIS AOD's. Based on figure 3 and table 2 and the discussion included, I do not see any comments on the impact of the differences exhibited, to the final results of your study. Comments are limited to: "The results discussed in this paragraph are comparable to the ones appearing in previous studies focusing on the Mediterranean region (see Papadimas et al., 2009; Koukouli et al., 2010). In general, it is shown here that the MODIS Terra Collection 051 data exhibit a better agreement with the ground-based observations from AERONET than MODIS Aqua data do."

We thank the Editor for giving us the opportunity to clarify this in the paper. We have added the following lines in the revised manuscript.

Page 18: "...Therefore, the statistics appearing for MODIS Terra throughout the paper could be considered more robust..."

Page 19: "...Only areas in Northern Africa are expected to be affected by the use of DB data due the extended lack of DT data there..."

3. So in the conclusions it would be important to include some discussion on this issue. For example figure 3c (DB) statistics are quite random in my opinion. So the importance of the results based on these data have to be discussed.

We thank the Editor for giving us the opportunity to clarify this. We have added the following lines in the first paragraph of the Summary and Conclusions section of the revised manuscript.

*"...According to the validation, the statistics appearing for MODIS Terra throughout the paper could be considered more robust while areas in Northern Africa are expected to be affected by the extended use of DB data which do not exhibit a very good matching with the ground-based observations..."*

4. Personally I do not like the function and function statistics of the articles vs years in figure 4. I think you can simply mention it or leave the line there for demonstrating what you want to say.

*The function has been removed from the Fig. 4 according to the Editor's recommendation.*

5. Figure 6. It was strange for me not to see any AOD hints in and around the area of Istanbul at all seasons. There are studies showing the importance of the aerosol flow from this area over various directions but in this study it is completely invisible e.g. compared to Ankara, Turkey area, or even Thessaloniki and Athens area.

We thank the Editor for his comment. We have double-checked our calculations following the Editor's recommendation. The same appears also in a very recent paper by Georgoulias et al. (2016)* where we compare the Collection 6 and 5.1 MODIS data. So, this has definitely not to do with the Collection used here. On the other hand the AODs are pretty high over the greater Istanbul region. This is more prominent in Fig. 11 (first column) where the anthropogenic aerosols are presented.

*Georgoulias, A. K., Alexandri, G., Kourtidis, K. A., Lelieveld, J., Zanis, P., and Amiridis, V.: Differences between the MODIS Collection 6 and 5.1 aerosol datasets over the greater Mediterranean region, Atmos. Environ., 147, 310-319, doi:10.1016/j.atmosenv.2016.10.014, 2016.*

6. Figure 7. FMF fraction use over land as the authors report is quite uncertain. Could you quantify the effect of its use on the land related percentages of this figure ?

As discussed in detail in the methodology we do not use FMF over land. We use the MACC-GOCART dust/total AOD ratios instead. Therefore our results do not suffer from the use of FMF.

7. Figure 11 dust cases/column. We expect a gradient of AOD based on the dust sources of Sahara and middle East. However e.g. figure 11j there are significant contributions in the order of 0.1 to 0.15 of AOD to specific isolated areas e.g. in Greece. Could it be due to uncertainties related with the way of characterizing dust here ?

Working with regional climate models in the region we have seen that generally local dust sources are not taken into account. Industrial, construction and agricultural activities are expected to produce significant amount of dust. For example in Greece, high dust concentrations appear over such regions (e.g. the industrial region in Thessaloniki, the Thessaly plain in central Greece, etc.).

**Main document changes and comments**

| Page 18: Inserted | User | 10/15/2016 12:50:00 PM |
|---|---|---|

Therefore, the statistics appearing for MODIS Terra throughout the paper could be considered more robust.

| Page 19: Inserted | User | 10/15/2016 12:52:00 PM |
|---|---|---|

 Only areas in Northern Africa are expected to be affected by the use of DB data due the extended lack of DT data there.

| Page 22: Inserted | User | 10/15/2016 1:49:00 PM |
|---|---|---|

; Georgoulias et al., 2016

| Page 22: Inserted | User | 10/15/2016 1:46:00 PM |
|---|---|---|

; Georgoulias et al., 2016

| Page 29: Inserted | User | 10/15/2016 12:56:00 PM |
|---|---|---|

According to the validation, the statistics appearing for MODIS Terra throughout the paper could be considered more robust while areas in Northern Africa are expected to be affected by the extended use of DB data which do not exhibit a very good matching with the ground-based observations.

| Page 38: Inserted | User | 10/15/2016 1:50:00 PM |
|---|---|---|

Georgoulias, A. K., Alexandri, G., Kourtidis, K. A., Lelieveld, J., Zanis, P., and Amiridis, V.: Differences between the MODIS Collection 6 and 5.1 aerosol datasets over the greater Mediterranean region, Atmos. Environ., 147, 310-319, doi:10.1016/j.atmosenv.2016.10.014, 2016.

| Page 52: Inserted | User | 10/15/2016 1:53:00 PM |
|---|---|---|

[Figure]

[Figure]

English (U.K.)

[Figure]

Header and footer changes

Text Box changes

Header and footer text box changes

Footnote changes

Endnote changes

[revised manuscript text omitted]